# Double-Filter: Efficient Fine-tuning of Pre-trained Vision-Language Models via Patch&Layer Filtering

**Yaoqin He** [* 1]  **Junchen Fu** [* 1]  **Kaiwen Zheng** [1]  **Songpei Xu** [1]  **Fuhai Chen** [2]  **Jie Li** [3]  **Joemon M. Jose** [1]  **Xuri Ge** [† 4]

## Abstract

In this paper, we present a novel approach, termed **Double-Filter**, to "slim down" the fine-tuning process of vision-language pre-trained (VLP) models via filtering redundancies in feature inputs and architectural components. We enhance the fine-tuning process using two approaches. First, we develop a new patch selection method incorporating image patch filtering through background and foreground separation, followed by a refined patch selection process. Second, we design a genetic algorithm to eliminate redundant fine-grained architecture layers, improving the efficiency and effectiveness of the model. The former makes patch selection semantics more comprehensive, improving inference efficiency while ensuring semantic representation. The latter's fine-grained layer filter removes architectural redundancy to the extent possible and mitigates the impact on performance. Experimental results demonstrate that the proposed Double-Filter achieves superior efficiency of model fine-tuning and maintains competitive performance compared with the advanced efficient fine-tuning methods on three downstream tasks, VQA, NLVR and Retrieval. In addition, it has been proven to be effective under METER and ViLT VLP models.

## 1. Introduction

Vision-Language Pre-trained (VLP) models (Du et al., 2022; Chen et al., 2020; Lu et al., 2019; Dou et al., 2022b) have established themselves as essential tools within the research community. Through large-scale pre-training on extensive

---

[*]Equal contribution [1]School of Computing Science, University of Glasgow, Glasgow, United Kingdom. [2]School of Computer and Data Science, Fuzhou University, Fuzhou, China. [3]School of Informatics, Xiamen University, Xiamen, China. [4]School of Artificial Intelligence, Shandong University, Jinan, China. Correspondence to: Xuri Ge <xuri.ge@sdu.edu.cn>.

*Proceedings of the $42^{nd}$ International Conference on Machine Learning*, Vancouver, Canada. PMLR 267, 2025. Copyright 2025 by the author(s).

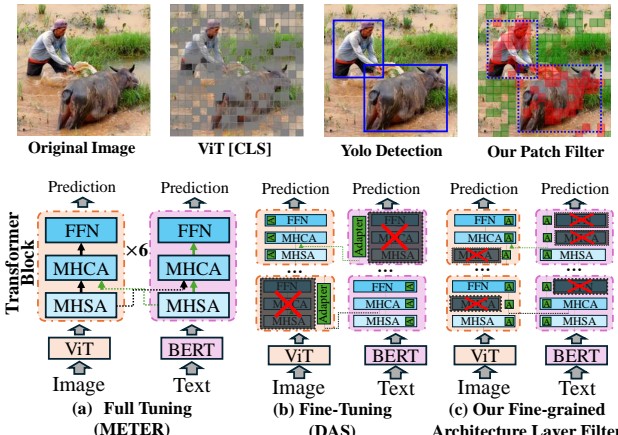

*Figure 1.* VLP models exhibit two primary types of redundancy: image patches and architectural layers. For the former, we segment patches into foreground and background by a YOLO detector, then filter them based on their importance as determined by a pre-trained ViT model. For the latter, fine-grained layer filtering overcomes architectural optimization bottlenecks, offering greater flexibility.

datasets (e.g., paired image-text datasets), these models achieve remarkable performance when fully fine-tuned for tasks such as visual question answering (VQA) (Zhou et al., 2020; Hong et al., 2021), natural language for visual reasoning (NLVR) (Suhr et al., 2019) and cross-modal retrieval (Gao et al., 2024; Long et al., 2024), etc. However, fully fine-tuning these models for downstream tasks incurs substantial computational costs, prompting researchers to investigate more efficient fine-tuning methods, such as integrating adapters or replacing specific components within the model's backbone.

Mainstream efficient fine-tuning methods, such as adapter-based and other parameter-efficient approaches (Houlsby et al., 2019; Hu et al., 2021; Karimi Mahabadi et al., 2021; Gao et al., 2024; Fu et al., 2024b), reduce the number of trainable parameters by incorporating adapter modules into the layers of the pre-trained backbone. Although these methods lower the training costs by freezing the backbone network, they are still inefficient in the inference process because they do not optimize the original architecture of the VLP model. Considering the architecture redundancies in the complex VLP-based backbone and extra parameters

of adapters, some recent works (Wu et al., 2024b; Kim et al., 2021; Wu et al., 2024a) focused on removing such redundancy in VLP models. For example, a simple block-skipping strategy (Wu et al., 2024b) reduces redundancy in VLP models by replacing complex blocks in the original backbone with adapters that minimally impact downstream tasks. These studies prove the existence of architectural redundancy in existing VLP models and alleviate the problem of inefficiency in the architecture level to a certain extent.

However, these basic, coarse-grained de-redundancy methods, which reduce transformer blocks in VLP models, overlook the finer architectural details, such as the sub-layers within transformer blocks. Specifically, as shown in Figure 1, when a transformer block is identified as redundant (as in (Wu et al., 2024b)), all its components (e.g., cross-attention, self-attention, and feed-forward network layers) are removed together, overlooking opportunities for selective optimization. Furthermore, these studies overlook data-level redundancy. In VLP models, both text and image data are processed; however, the main bottleneck arises from image patches, as their quantity significantly exceeds the length of text in these tasks (see Section 3.3 for details). As shown in MAE (He et al., 2022b), an image can be effectively reconstructed using just 25% of its patches, indicating that many patches are redundant and hinder model efficiency. However, directly applying the random patch selection strategy from (He et al., 2022b) risks omitting valuable image information, leading to significant information loss. (Liang et al., 2022) addresses this by using the ViT (Alexey, 2020) classification token to rank and select patches with the highest scores, however, this approach mainly captures objects and often neglects contextual information essential for complex multimodal tasks.

To address these two issues, we propose a novel **Double-Filter** to address both data-level and architecture-level redundancies, enabling efficient fine-tuning of VLP models.

**Novel Data-level Redundancy Removal:** We introduce an *Image Patch Filter* (IPF) to eliminate extraneous image patches that do not impact contextual semantics. Specifically, using an offline YOLO detector (Redmon, 2016), we distinguish between foreground and background areas, where detected salient objects represent the foreground, and the remaining areas are considered background. Low-importance patches within both foreground and background regions are then filtered based on the attention scores of the classification ([CLS]) token. As illustrated in Figure 1, our IPF effectively preserves global semantics across patches, rather than focusing solely on salient objects.

**Novel Architecture-level Redundancy Removal:** At the architectural level, we introduce a novel adjustable genetic algorithm (AGA) to precisely identify and filter redundant fine-grained sub-layers within the transformer blocks of

VLP models, and implemented in our *fine-grained Architecture Layer Filter* (ALF). Unlike traditional methods (Wu et al., 2024b) that replace entire transformer blocks, our approach targets specific sub-components, such as the Multi-Head Self-Attention layer (MHSA), Multi-Head Cross Attention layer (MHCA), and Feed-Forward Network (FFN) layer. By selectively filtering these fine-grained sub-layers and substituting them with lightweight adapters, we effectively reduce architecture-level redundancy at a more granular level. Notably, our AGA evaluates the filtering effects of complex sub-layer combinations through multi-generation crossover inheritance and regulatory mutation, guided by a specially designed fitness function.

Our contributions in this paper are as follows:

- We propose a novel multi-level efficient fine-tuning strategy for VLP models, called Double-Filter, by removing both image patch redundancy and fine-grained architecture layer redundancy.

- We devise a new *Image Patch Filter* (IPF) that filters redundant patches using ViT attentions, applied separately to the foreground and background as detected by YOLO. This approach ensures both spatial and semantic integrity.

- We design an efficient adjustable genetic algorithm-based (AGA) fine-grained *Architecture Layer Filter* (ALF) designed to minimize architectural layer redundancy in VLP models for downstream fine-tuning. ALF leverages multi-generation crossover inheritance and regulatory mutation to swiftly identify filtered model variants, eliminating redundant sublayers.

Experimental results show that by filtering over **60%** of image patches and **12** model layers, our Double-Filter achieves comparable performance on METER model fine-turning compared to state-of-the-art PEFT methods, reducing computational costs by **21.18G** in terms of FLOPs. In addition, Double-filler is equally effective in the ViLT model.

## 2. Related Work

**Vision Language Pre-trained models** Vision-language pre-training (VLP) (Chen et al., 2024; Du et al., 2022) has advanced through leveraging large-scale image-text pairs for tasks such as masked language modeling, masked image modeling, and Image-Text Matching. Most VLP models such as ViL-BERT (Lu et al., 2019) and LXMERT (Tan & Bansal, 2019) relied on separate encoders for visual and textual features (e.g., BERT for text, Faster-RCNN for images) and used independent Transformer branches for cross-modal interaction. Later models like VisualBERT (Li et al., 2019), VL-BERT (Su et al., 2019), and UNITER simplified this by integrating both modalities into a single Transformer network. METER (Dou et al., 2022b) takes this a

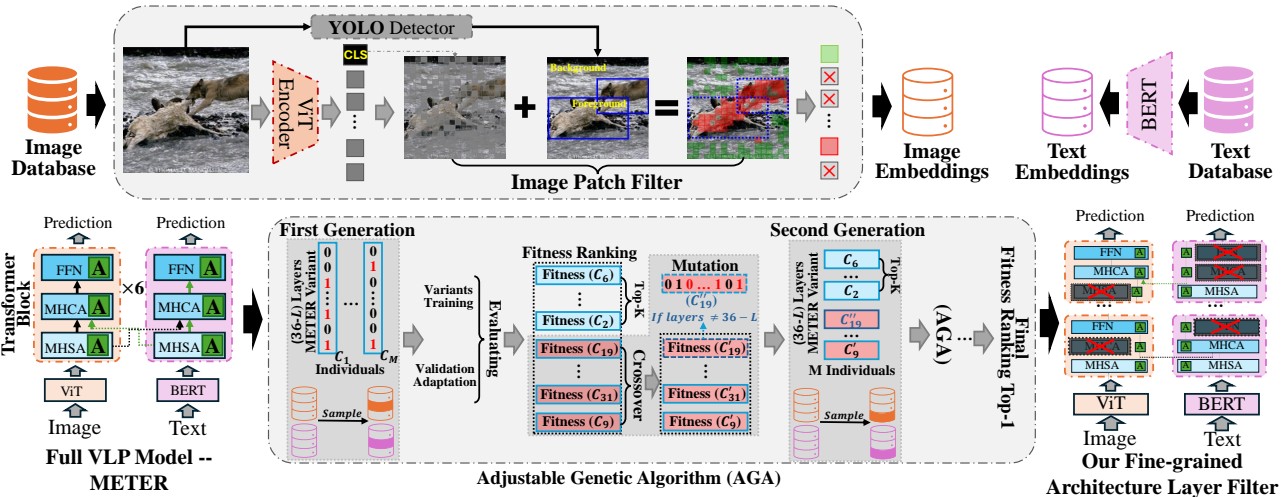

*Figure 2.* The proposed Double-Filter framework comprises two primary components: the **Image Patch Filter** (IPF) and the **fine-grained Architecture Layer Filter** (ALF). The IPF utilizes a YOLO detector to partition an image's foreground and background areas and then selects patches based on their [CLS] weights in each area. The Fine-grained ALF introduces new Adjustable Genetic Algorithm (AGA), which formulates layer filtering as a genetic algorithm optimization problem to approximate optimal fine-grained filtering across layers.

step further by fully embracing an end-to-end transformer-based design, exploring various combinations of vision encoders (CLIP-ViT (Radford et al., 2021)) and text encoders (RoBERTga (Liu et al., 2019), DeBERTa (He et al., 2020)), while also optimizing multimodal fusion mechanisms (e.g., merged attention vs. co-attention). Despite these improvements, many VLP models still exhibit redundancy during the finetuning at both model and data levels, leading to inefficiencies. The approach proposed in this paper aims to deliver a universal and efficient fine-tuning solution that addresses these challenges across a broad spectrum of VLP models, particularly those with complex cross-modal interactions.

**Efficient Finetuning** Previous studies have largely focused on adapter-based methods (Gao et al., 2024; Sung et al., 2022; Zhang et al., 2022; Feng et al., 2025; Zanella & Ben Ayed, 2024; Wu et al., 2023; 2024a; Fu et al., 2024a) to enable efficient fine-tuning of Vision-Language Pre-trained (VLP) models, aiming to lower the adaptation cost to downstream tasks. However, these approaches often introduce architectural redundancy due to the addition of new trainable modules, which can limit overall efficiency. To tackle these challenges, recent skipping-based approaches like DAS (Wu et al., 2024b) have introduced parameter and computation efficient transfer learning (PCETL). However, this approach primarily targets structural redundancy by replacing entire transformer blocks at a coarse level, overlooking a crucial efficiency bottleneck—patch redundancies in visual data and finer-grained architectural layers. In contrast, the Double Filtering framework uniquely addresses both architectural and patch-level redundancies simultaneously, distinguishing

it from prior research approaches.

# 3. The Proposed Approach: Double-Filter

Figure 2 provides an overview of our **Double-Filter** method, applied to the widely used VLP model METER (Dou et al., 2022b). The method consists of two main components: the *Image Patch Filter* and the *Fine-Grained Architecture Layer Filter*. Double-Filter is designed to facilitate comprehensive and efficient fine-tuning at both the patch and architectural levels in VLP models, particularly those with complex cross-modal interactions.

In Section 3.1, we outline the image patch selection process, detailing the foreground-background token filtering based on the attention scores for the [CLS] token from Vision Transformers (ViT) (Alexey, 2020). Section 3.2 discusses fine-grained architectural redundancy removal using our genetic algorithm, which prunes unnecessary layers while preserving model performance.

## 3.1. Image Patch Filter

In this section, we introduce the Image Patch Filter (IPF), a method designed to efficiently filter image patches by discarding less relevant regions while preserving vital background details that offer essential contextual cues. The innovation in IPF lies in its approach to partitioning the entire image into foreground and background portions using a YOLO detector, which enhances the informativeness and efficiency of the patch selection process.

**Algorithm 1** Image Patch Filter

1: **Input:** YOLO detector $\mathcal{D}$, Image Sparsity $\rho$
2: **Output:** Retained patches $T_{\text{PF}}^I$
3: Extract classification score $s_{\text{CLS}}$ for $I$;
4: Get $n$ bounding boxes $\{B_1, B_2, \ldots, B_n\} = \mathcal{D}(I)$
5: Get foreground $FG = \bigcup_{i=1}^{n} B_i$, Get background $BG = I - FG$
6: Compute the space occupancy ratio using the areas of $FG$ and $BG$: $\epsilon = \frac{\mathcal{A}(FG)}{\mathcal{A}(FG)+\mathcal{A}(BG)}$;
7: Compute the total number of patches to retain, $|T_{\text{PF}}^I| = \lceil(1 - \rho) \cdot |T^I|\rceil$;
8: Compute the number of patches for $FG$ and $BG$: $|T_{\text{PF}}^{FG}| = \lceil\epsilon \cdot |T_{\text{PF}}^I|\rceil$ and $|T_{\text{PF}}^{BG}| = |T_{\text{PF}}^I| - |T_{\text{PF}}^{FG}|$;
9: Select the top $|T_{\text{PF}}^{FG}|$ patches for $FG$ based on $s_{\text{CLS}}$: $T_{\text{PF}}^{FG} = \text{Top-}|T_{\text{PF}}^{FG}|\left(T^{FG}\right)$;
10: Select the top $|T_{\text{PF}}^{BG}|$ patches for $BG$ based on $s_{\text{CLS}}$: $T_{\text{PF}}^{BG} = \text{Top-}|T_{\text{PF}}^{BG}|\left(T^{BG}\right)$;
11: Obtain the retained patches: $T_{\text{PF}}^I = T_{\text{PF}}^{FG} \cup T_{\text{PF}}^{BG}$;

### 3.1.1. PATCH IMPORTANCE

VLP models typically represent an image as a sequence of patches (Alexey, 2020; Dou et al., 2022b;a). To enable effective patch filtering, we assign an importance score to each patch by leveraging the classification token ([CLS]) of the pre-trained ViT following (Liang et al., 2022).

The ViT discerns and weighs the relevance of various image patches, primarily through attention scores assigned by the [CLS] capturing their relative importance for image classification tasks. This attention mechanism can be formulated as follows:

$$s_{\text{CLS}} = \text{Softmax}\left(q_{\text{class}} \cdot \frac{K^T}{\sqrt{d}}\right)$$

where $q_{\text{class}}$ is the query vector associated with the [CLS] token. $K$ is the aggregated key matrices from all tokens. $d$ is the dimensionality of the query vector. $s_{\text{CLS}}$ denotes the attention weights that modulate the contributions of different tokens based on their computed relevance. In the multi-head attention layer, multiple sets of attention scores are computed for the [CLS] token, each reflecting different facets of the image's context.

### 3.1.2. FOREGROUND AND BACKGROUND

While filtering patches based on their importance appears logical, relying solely on the attention scores for [CLS] token tends to prioritize object-specific information, potentially overlooking essential background details necessary for complex multimodal tasks. To address this, we propose categorizing images into **foreground** and **background** portions.

**Algorithm 2** Fine-Grained Architecture Layer Filter

1: **Input:** Number of transformer blocks $N$, target number of layers $L$, number of elite chromosomes $K$, number of generations $G$, population size $M$, training set $TS$, validation set $VS$, fitness score function, crossover and mutation operations.
2: **Output:** Optimized architecture $C^*$ with reduced number of layers $L$.
3: Initialize $M$ chromosomes $C_m = [c_1^{(m)}, \ldots, c_{3N}^{(m)}]$, where $c_i^{(m)} \in \{0, 1\}$, for $m = 1$ to $M$, to represent architectural choices. Denote the population as $MC = \{C_1, C_2, \ldots, C_M\}$.
4: **for** $g = 1$ to $G$ **do**
5:    Sample random training and validation subsets: $\widetilde{TS} \subset TS, \widetilde{VS} \subset VS$.
6:    **for** $m = 1$ to $M$ **do**
7:       Train $C_m$ on $\widetilde{TS}$.
8:       Compute $FS_m = \text{Fitness}(C_m, \widetilde{TS}, \widetilde{VS})$.
9:    **end for**
10:   Compute $P_i = \frac{FS_i}{\sum_{j=1}^{M} FS_j}$ for each $C_i \in MC$.
11:   Select $K$ elite chromosomes with highest fitness scores: $EC = \text{Top-}K(MC)$.
12:   Get remaining chromosomes $RC = MC \setminus EC$
13:   Perform selection by sampling with a roulette wheel approach (Lipowski & Lipowska, 2012), choosing $|RC|$ parent chromosomes from $RC$ to form the set $PC$ based on $P$: $PC = \{C_1, C_2, \ldots, C_n \mid C_i \in RC\}$.
14:   Perform crossover to create child chromosomes: $CRC = \text{Crossover}(PC)$.
15:   **for** each $C_i'$ in $CRC$ **do**
16:       **if** $\sum_{j=1}^{3N} c_j^{(i)} \neq L$ **then**
17:          Perform mutation: $C_i'' = \text{Mutation}(C_i')$.
18:       **end if**
19:   **end for**
20:   Update $MC$ combining them with the elite chromosomes $EC$: $MC = CRC \bigcup EC$.
21: **end for**
22: **Return:** Optimized architecture $C^*$, which is the chromosome in $MC$ with the highest fitness score.

Specifically, for a given image $I$, its foreground and background are denoted as $FG$ and $BG$. The patches corresponding to $I$, $FG$, and $BG$ are denoted as $T^I$, $T^{FG}$, and $T^{BG}$, respectively. A YOLO Detector is employed to generate $n$ bounding boxes $\{B_i, B_2, \ldots, B_n\}$ for the image, where each bounding box represents an individual object. The foreground, $FG$, is defined as $FG = \bigcup_{i=1}^{n} B_i$, representing the union of these bounding boxes. The background, $BG$, comprises the remaining portions of the image, as $BG = I - FG$. [1]

The objective of IPF is to retain informative patches $T_{\text{PF}}^I$, with a sparsity $\rho \in (0, 1)$ compared to the original patches

---

[1] In practice, some patches overlap between $FG$ and $BG$; We also include these patches to $T^{FG}$ for simplicity.

$T^I$. Formally, $|T_{\text{PF}}^I| = \lceil \rho \cdot |T^I| \rceil$, where $|T|$ denotes the number of patches $T$, and $\lceil \cdot \rceil$ represents the ceiling operation. Once the target patch count is determined, a *space occupancy ratio* $\epsilon$ is introduced to specify the number of foreground patches $T_{\text{PF}}^{FG}$ and background patches $T_{\text{PF}}^{BG}$ to retain. The $\epsilon$ is calculated as the ratio of the area of the foreground regions to the area of the entire image. The number of retained foreground patches is calculated as $|T_{\text{PF}}^{FG}| = \lceil \epsilon \cdot |T_{\text{PF}}^I| \rceil$, while the number of retained background patches is defined as $|T_{\text{PF}}^{BG}| = |T_{\text{PF}}^I| - |T_{\text{PF}}^{FG}|$. Within each region, patches are ranked based on their importance scores, $s_{\text{CLS}}$, and only those with the highest scores are retained according to the defined counts. The detailed procedure for IPF processing is provided in Algorithm 1.

## 3.2. Fine-grained Architecture-Layer Filter

To mitigate redundancy in Transformer-based models for cross-modal tasks, we propose an Adjustable Genetic Algorithm (AGA) for fine-grained architecture filtering, approximating an optimal configuration [2]. For a fine-grained replacement, AGA decomposes Transformer blocks into detailed components, including Multi-Head Self-Attention (MHSA), Feed-Forward Network (FFN), and in some cases, Multi-Head Cross-Attention (MHCA), collectively termed "architecture layers (AL)". AGA selectively substitutes these computationally intensive ALs with lightweight adapter blocks (Houlsby et al., 2019). The novelty of AGA lies in three main aspects: (1) fine-grained architectural replacements; (2) novel formulation of architecture filtering task as a genetic algorithm; (3) We customize the design of mutation operation to satisfy the requirement while diversifying each generation effectively.

**Problem Formulation:** Specifically, for a VLP model with $N$ Transformer blocks, where each layer contains MHSA, MHCA, and FFN modules, each individual is encoded as a chromosome $C = [c_1, c_2, \ldots, c_{3N}]$. Here, each gene $c_i \in \{0, 1\}$ indicates whether the corresponding layer is replaced by a lightweight adapter: a value of 1 indicates replacement while 0 signifies retention of the original structure.

The AGA aims to identify the approximate optimal reduction with $L$ filtered layers. AGA proceeds over $G$ generations, with $M$ individuals in each generation. In each generation, individuals are evaluated on subsets $\widetilde{TS}$ and $\widetilde{VS}$ that are randomly sampled from the training set $TS$ and validation set $VS$, respectively. The most promising individuals from the final generation is selected as the final architecture. Similar to traditional genetic algorithms (Sampson, 1976; Mirjalili & Mirjalili, 2019), we define our fitness function, crossover, and mutation operations as follows.

[2]Note that identifying an absolute optimal structure is NP-hard, rendering exhaustive search impractical.

**Fitness Function:** The fitness function is crucial in a genetic algorithm, evaluating each individual's quality. In AGA, the fitness function is defined as:

$$\text{Fitness}(C, \widetilde{TS}, \widetilde{VS}) = -\frac{1}{2}\left(\mathcal{L}(C, \widetilde{TS}) + \mathcal{L}(C, \widetilde{VS})\right) + \beta,$$

where $\mathcal{L}(\cdot, \cdot)$ denotes the loss on sampled datasets for a given model, and a bias term $\beta$ is defined as $\beta > \frac{1}{2}\max\left(\left\{\mathcal{L}(C_i, \widetilde{TS}) + \mathcal{L}(C_i, \widetilde{VS}) \mid i \in \{1, 2, \cdots, M\}\right\}\right)$ to ensure that the Fitness score remains positive for the following selection.

**Crossover Operation:** The crossover operation generates fresh individuals through single-point or two-point crossover methods. In the single-point crossover method, a random crossover point $k$ is selected, and genes beyond this point are swapped between two parent individuals, producing two offspring individuals:

$$C_{\text{child1}}' = [c_1^E, \ldots, c_k^E, c_{k+1}^F, \ldots, c_{3N}^F];$$

$$C_{\text{child2}}' = [c_1^F, \ldots, c_k^F, c_{k+1}^E, \ldots, c_{3N}^E],$$

where $E$ and $F$ represent the two parent individuals.

**Mutation Operation:** To cater to the replacement of $L$ layers within the chromosome $C'$, we propose a mutation operation for adjusting gene values. Firstly, we calculate $Z = \sum_{i=1}^{3N} c_i$ which gives the current count of genes with value 1. If $Z \neq L$, we then adjust as follows: if $Z > L$, randomly select $Z - L$ genes with a value of 1 and switch them to 0; if $Z < L$, select $L - Z$ genes with a value of 0 and change them to 1. This mutation process ensures that the updated chromosome $C''$ fulfills the condition of $\sum_{i=1}^{3N} c_i = L$, ensuring the desired $L$ layers are replaced.

The Double-Filter strategy combines an IPF strategy for image patch selection with an AGA for a Fine-grained Architecture-Layer Filter, comprehensively enhancing the efficiency of fine-tuning for VLP models.

## 3.3. FLOPs Analysis

In this section, we analyze the Floating Point Operations (FLOPs) (Li et al., 2020; Hobbhahn & Sevilla, 2021) required by the transformer model. Using the image tower from the fusion layer of METER (Dou et al., 2022b) as an example, we theoretically estimate the potential FLOPs savings offered by the Double-Filter Framework compared to the vanilla model.

We denote the number of transformer blocks in METER (Dou et al., 2022b), the sequence length of the image, the sequence length of the text, and the hidden dimension as $N$, $S$, $\widetilde{S}$, and $D$, respectively. Note that, in practical VLP models, $\widetilde{S} \ll S$, and we consider a batch size of 1 for simplicity. Within a transformer block, the main computational

costs arise from the Multi-Head Self-Attention (MHSA), Multi-Head Cross-Attention (MHCA), and Feed-Forward Network (FFN). Here, we focus our analysis on the FLOPs associated with these three components.

**FLOPs in Multi-Head Self-Attention (MHSA)**: Each MHSA layer consists of three projection operations ($6SD^2$), Query, Key, and Value, followed by the computation of attention scores ($2S^2D$), application of these scores ($2S^2D$), and a subsequent linear projection of the concatenated heads ($2SD^2$).[3] Therefore, the total FLOPs for MHSA can be expressed as:

$$\text{FLOPs}_{\text{MHSA}} = 6SD^2 + 2S^2D + 2S^2D + 2SD^2$$
$$= 8SD^2 + 4S^2D$$

**FLOPs in Multi-Head Cross-Attention (MHCA):** The primary distinction between MHCA and MHSA lies in the sequence length of the Query, Key, and Value. The Query has the same sequence length as the image, while the Key and Value align with the sequence length of the text. Consequently, the FLOPs can be formulated as:

$$\text{FLOPs}_{\text{MHCA}} = (2SD^2 + 4\widetilde{S}D^2) + 2S\widetilde{S}D + 2S\widetilde{S}D + 2SD^2$$
$$= 4SD^2 + 4\widetilde{S}D^2 + 4S\widetilde{S}D$$
$$\approx 4SD^2$$

We omit the term $4\widetilde{S}D^2 + 4S\widetilde{S}D$ since $\widetilde{S} \ll S$, allowing for a clearer computation in the following steps.

**FLOPs in Feed-Forward Network (FFN)**: A standard FFN has an expansion factor of 4, meaning the hidden layer size is $4D$. Consequently, the total FLOPs for the FFN is $16SD^2$.

**Total FLOPs for a transformer model**: Therefore, the total FLOPs for $N$ transformer blocks are:

$$\text{FLOPs}_{\text{total}} = N(\text{FLOPs}_{\text{MHSA}} + \text{FLOPs}_{\text{MHCA}} + \text{FLOPs}_{\text{FFN}})$$
$$\approx N\left((8SD^2 + 4S^2D) + 4SD^2 + 16SD^2\right)$$
$$\approx 28NSD^2 + 4NS^2D$$

**FLOPs with Double-Filter**: Using the Double-Filter framework, we retain a fraction $p$ of the original patches (sequence length), $q$ in the MHSA layers, and $r$ in the MHCA, and $o$ in FFN, where $0 < p, q, r, o < 1$. The total FLOPs under the Double-Filter (DF) can then be calculated as:

$$\text{FLOPs}_{\text{DF}} \approx N\left(q(8pSD^2 + 4pS^2D) + 4rSD^2 + 16poSD^2\right)$$
$$\approx (8pq + 16po + 4r)NSD^2 + 4pqNS^2D$$

This analysis demonstrates that the Double-Filter Framework can reduce the FLOPs of a transformer model from

$28NSD^2 + 4NS^2D$ to approximately $(8pq + 16po + 4r)NSD^2 + 4pqNS^2D$. In practical computation, we also consider incorporating the FLOPs of YOLO; however, since its computational complexity is significantly smaller compared to the transformer model, we omit it from the theoretical analysis.

# 4. Experiment

In this section, we conduct experiments to evaluate the proposed Double-Filter for efficient fine-tuning of VLP models.

## 4.1. Datasets & Metrics

We validated our approach on three visual language tasks:

**Visual Question Answering (VQA):** We conducted experiments on the VQA2.0 dataset (Goyal et al., 2017), which transforms traditional open-ended natural language questioning into a multi-classification task with 3,129 answer options. Following the foundational configurations of the METER (Dou et al., 2022b) and ViLT (Kim et al., 2021) models, we trained the models on the training and validation sets and assessed performance through online evaluations of the test set.

**Natural Language for Visual Reasoning (NLVR):** NLVR²(Suhr et al., 2019) dataset is designed as a binary classification to determine whether a text accurately describes the content of two images. According to the default settings of ViLT and METER, we first input one of the images combined with the text into the model, followed by the other image combined with the same text. The final prediction is derived by concatenating the outputs from these two inputs.

**Cross-modal Retrieval:** We evaluate the cross-modal retrieval capability of our model on the Flickr30K dataset (Plummer et al., 2015), using the standard split introduced by (Karpathy & Fei-Fei, 2015; Ge et al., 2024). For initialization, we follow the same operation as DAS (Wu et al., 2024a), leveraging the pre-trained retrieval heads of METER and ViLT for similarity estimation. To improve the robustness of training, we adopt a hard-negative mining strategy where each positive sample is paired with 15 randomly sampled negatives per iteration.

**Metric:** We evaluate using accuracy for NLVR and VQA, and Recall@1 for the image retrieval (IR) and text retrieval (TR).

## 4.2. Implementation Details

We conducted experiments on the METER (Dou et al., 2022b) and ViLT (Kim et al., 2021) models. For the ME-

---

[3]The FLOPs for matrix multiplication $[m, n] \times [n, p]$ is $2mnp$ (Ozaki et al., 2011).

*Table 1.* Comparison between **Double-Filter** and other PEFT methods for widely used VLP modals, METER (Dou et al., 2022b) and ViLT (Kim et al., 2021), on VQA and NLVR tasks, as well as cross-modal retrieval task. The best performance is **bold** and the second best is underlined.

| METER | | | | | | | |
|---|---|---|---|---|---|---|---|
| **Method** | **Updated Param.** | **VQA** | | **NLVR** | | **Retrieval** | |
| | | **test-dev** | **+FLOPs** | **test-P** | **+FLOPs** | **IR/TR R@1** | **+FLOPs** |
| Full Tuning | 323.31M | 77.43 | 0.00 | 83.05 | 0.00 | 82.22/94.30 | 0.00 |
| Classifier Only | - | 69.93 | 0.00 | 73.23 | 0.00 | 78.80/89.00 | 0.00 |
| Shallow Prompt | 0.30M | 68.51 | +28.71G | 65.69 | +26.84G | 74.20/88.60 | +28.71G |
| Deep Prompt | 1.84M | 70.78 | +6.53G | 72.64 | +5.59G | 78.84/89.40 | +6.53G |
| LoRA | 0.29M | 74.00 | 0.00 | 78.82 | 0.00 | 79.86/92.60 | 0.00 |
| Adapter | 5.34M | 74.70 | +1.64G | 79.93 | +1.38G | 80.38/91.90 | +1.64G |
| Scaled PA | 3.59M | **75.11** | +1.12G | **80.38** | +0.66G | **80.40/93.20** | +1.12G |
| DAS | 5.34M | 74.80 | -11.16G | 80.11 | -5.13G | 80.12/91.80 | -11.16G |
| **Double-Filter** | 5.34M | 74.25 | **-21.18G** | 80.12 | **-12.51G** | 80.05/91.22 | **-21.18G** |

| ViLT | | | | | | | |
|---|---|---|---|---|---|---|---|
| **Method** | **Updated Param.** | **VQA** | | **NLVR** | | **Retrieval** | |
| | | **test-dev** | **+FLOPs** | **test-P** | **+FLOPs** | **IR/TR R@1** | **+FLOPs** |
| Full Tuning | 115.43M | 71.26 | 0.00 | 76.13 | 0.00 | 64.40/83.50 | 0.00 |
| Classifier Only | - | 65.75 | 0.00 | 66.08 | 0.00 | 57.42/78.00 | 0.00 |
| Shallow Prompt | 0.15M | 66.47 | +19.53G | 66.47 | +19.53G | 55.92/74.80 | +19.53G |
| Deep Prompt | 1.84M | 69.30 | +5.14G | 73.34 | +5.14G | 58.64/79.50 | +5.14G |
| LoRA | 0.15M | 68.44 | 0.00 | 72.77 | 0.00 | 57.44/77.70 | 0.00 |
| Adapter | 3.56M | **70.85** | +0.86G | **75.51** | +0.86G | **62.68/81.40** | +0.86G |
| Scaled PA | 1.80M | 70.40 | +0.44G | 75.13 | +0.44G | 61.88/79.00 | +0.44G |
| DAS | 3.56M | 69.28 | -1.03G | 74.89 | -1.03G | 60.66/80.80 | -1.03G |
| **Double-Filter** | 3.56M | 68.37 | **-4.72G** | 74.50 | **-4.72G** | 61.18/79.39 | -4.72G |

TER model, we focus on filtering the most computationally intensive component: the fusion layer for METER and, the transformer encoder for ViLT. The adapter bottleneck for both networks was set to 96 following (Wu et al., 2024b). We used YOLOv8 (Reis et al., 2023) as the Yolo Detecter. Notably, compared with METER, ViLT does not include the fusion network. So we only consider de-redundancy of its encoders. Therefore, when calculating FLOPs cost in the later experiments, we only considered YOLO's FLOPs for METER, not ViLT. The FLOPs of YOLO are about 4.5G. For images without any detected objects, we used the median of CLS scores to distinguish between foreground and background. To better align the IPF method with the ViLT model, we did not skip the first block's self-attention layer, and other settings remained consistent with the base settings of the METER and ViLT models. The other detailed settings for Double-Filter are placed in Appendix Table 1. All experiments were conducted on one RTX 3090 GPU.

### 4.3. Efficiency-Performance Comparison

We evaluate our proposed **Double-Filter** based on two widely-used VLP models, i.e. METER (Dou et al., 2022b) and ViLT (Kim et al., 2021), on three benchmarks (NLVR$^2$ (Suhr et al., 2019) , VQA (Goyal et al., 2017) and Flickr30K (Plummer et al., 2015)) of different downstream tasks. We compare it on efficiency and effective fine-tuning with other popular PEFT methods, including Shallow Prompt (Li &

*Table 2.* Efficient comparisons of different methods in terms of inference time (items/s) on two tasks based on different VLP models with the inference time.

| METER | | | | |
|---|---|---|---|---|
| **Method** | Full Tuning | Adapter | DAS | **Double-Filter** |
| **VQA** | 55.32 | 54.05 | 60.49 | **70.72** |
| **NLVR** | 46.50 | 45.73 | 51.47 | **55.13** |
| ViLT | | | | |
| **VQA** | 115.50 | 113.62 | 124.63 | **135.24** |
| **NLVR** | 54.49 | 53.86 | 55.80 | **58.36** |

Liang, 2021), Deep Prompt (Jia et al., 2022), LoRA (Hu et al., 2021), Adapter (Sung et al., 2022), Scaled PA (He et al., 2022a), and DAS (Wu et al., 2024b).

Detailed comparisons are shown in Table 1. Existing PEFT methods generally use fewer updated parameters compared to full finetuning. However, except for our Double-Filter, DAS, and LoRA, other methods tend to increase computational complexity (e.g. FLOPs). While LoRA does not increase computational complexity, its performance is lower than the other two methods. Specifically, compared to methods with lower inference complexity, our Double-Filter achieves the largest reduction, with an average of more than twice the reduction compared to DAS (-21.18G FLOPs for VQA, -12.51G FLOPs for NLVR on METER). Moreover, the performance degradation is minimal, averaging only 0.30% on the METER network and 0.55% on ViLT network. We also compared full tuning, Adapter, DAS, and our

*Table 3.* Efficiency-Performance balance of **Image Patch Filter (IPF)** for METER and ViLT on VQA, NLVR tasks.

| Method | Patch Filtering | VQA | | NLVR | |
|---|---|---|---|---|---|
| | | test-dev | +FLOPs | test-P | +FLOPs |
| **METER** | | | | | |
| Baseline | 0 | 75.28 | 1.68G | 81.28 | 0.99G |
| IPF | 40% | 75.16 | -8.97G | 81.15 | -4.49G |
| | 50% | 75.01 | -14.81G | 81.16 | -6.59G |
| | 60% | 74.67 | -16.52G | 81.32 | -8.6G |
| | 70% | 74.16 | -20.24G | 80.14 | -10.6G |
| **ViLT** | | | | | |
| Baseline | 0 | 70.13 | 0.73G | 76.26 | 0.73G |
| IPF | 20% | 69.32 | -3.06G | 75.47 | -3.06G |
| | 30% | 68.95 | -5.09G | 74.68 | -5.09G |
| | 40% | 68.36 | -7.11G | 74.60 | -7.11G |

*Table 4.* Efficiency-Performance balance of fine-grained architecture layer filter for METER and ViLT on VQA and NLVR tasks.

| Method | Filtering Layers | VQA | | NLVR | |
|---|---|---|---|---|---|
| | | test-dev | +FLOPs | test-P | +FLOPs |
| **METER** | | | | | |
| Baseline | 0 | 75.28 | 1.71G | 81.28 | 1.02G |
| ALF | 6 | 74.92 | -3.84G | 80.60 | -2.64G |
| | 12 | 74.82 | -8.43G | 80.98 | -7.57G |
| | 18 | 74.23 | -22.9G | 80.06 | -10.03G |
| | 24 | 73.61 | -27.91G | 80.25 | -14.25G |
| **ViLT** | | | | | |
| Baseline | 0 | 70.13 | 0.73G | 76.26 | 0.73G |
| ALF | 2 | 69.15 | -1.04G | 75.50 | -1.04G |
| | 4 | 67.78 | -3.06G | 75.15 | -3.59G |
| | 6 | 66.63 | -5.08G | 73.88 | -5.6G |

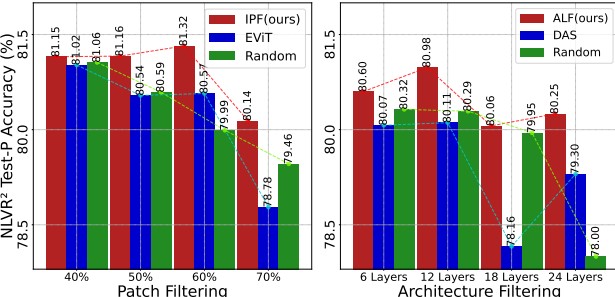

*Figure 3.* Ablation studies of different image patch filtering methods, i.e. **Image Patch Filter (IPF)** vs. EViT (Liang et al., 2022) vs. Random Filter, and architecture filtering methods, i.e. fine-grained architecture layer filter (ALF) vs. DAS (Wu et al., 2024b) vs. Random Filter, for METER on NLVR task.

method regarding inference speed. In Table 2, our proposed Double-Filter improves the inference speed, demonstrating superior efficiency. Our proposed Double-Filter exhibits the lowest computational complexity among all existing PEFT methods, improving inference speed. Additionally, it shows competitive results compared to other PEFT methods.

## 4.4. Efficiency-Performance Balance of Image Patch Filter

We conduct extensive ablation studies to investigate the impact of different variants of the proposed image patch filter (IPF) on performance and efficiency. We investigate the effects of different patch filtering ratios and different filtering methods in Table 3 and Figure 3. Specifically, in VQA task of the METER model (Dou et al., 2022b), as the patch filtering ratio increases, although the accuracy decreases slightly, the FLOPs decrease significantly in Table 3. It shows that our IPF greatly helps model fine-tuning efficiency. Additionally in the NLVR$^2$ task, reducing different patches can sometimes improve performance, suggesting that redundant patches may have a negative impact on performance.

In the left of Figure 3, to further validate the effectiveness of our proposed IPF, we conducted ablation studies comparing

random patch filtering (**Random**) and patch filtering based on CLS attention scores (**EViT** (Liang et al., 2022)). The experiments are performed on the NLVR task on METER model. The results show that our IPF can better maintain performance when filtering patch redundancy. And this advantage becomes greater as the patch filtering rate increases. At a reasonable ratio, we can achieve the best efficiency and performance balance for VLP model fine-tuning.

## 4.5. Efficiency-Performance Balance of Architecture Layer Filter

In this section, we conducted ablation experiments on our proposed ALF method by skipping different layers, and observed that the METER model exhibits a significant amount of redundancy. For instance, even after removing 24 layers out of a total of 36 layers in the NLVR task, it still performs well. In contrast, ViLT shows a noticeable performance decline after removing just 4 layers. This discrepancy likely stems from the structural differences between the models: METER employs two separate encoders for processing textual and visual information and includes cross-attention modules, whereas ViLT only utilizes word embeddings and image patch embeddings, and lacks cross-attention modules.

To further validate the effectiveness of our ALF method, we compare it against DAS (Wu et al., 2024b) and a Random approach (which randomly removes an equivalent number of layers) in the NLVR task on the METER model. As shown in the right of Figure 3, compared to two other methods, our ALF effectively maintains performance stability even with more layers removed, demonstrating its advantage in balancing model reduction and performance retention.

## 4.6. Visualization

To more intuitively demonstrate our proposed **Double-Filter** method, we provide detailed visualizations of the results of Image Patch Filter (IPF) and fine-grained Architecture Layer Filter (ALF) in Figure 4. It shows that our IPF effectively preserves key information in images. Compared

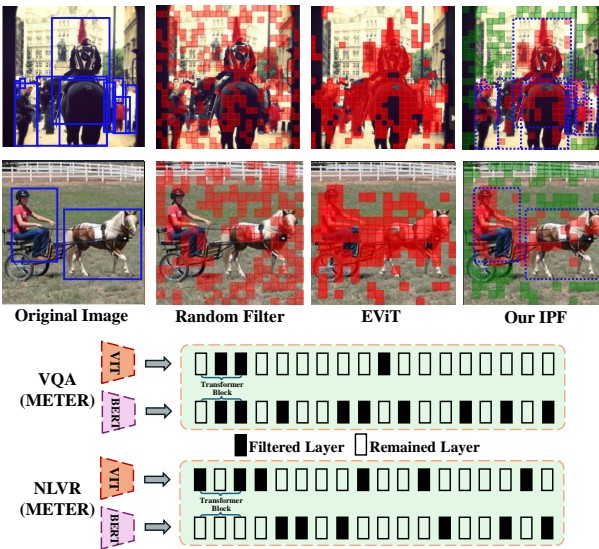

*Figure 4.* Visualizations of the proposed Double-Filter for METER (Dou et al., 2022b) on NLVR and VQA tasks, including the results of our proposed image patch filter and fine-grained architecture layer filter.

with EViT (Liang et al., 2022) which only focuses on salient objects, our IPF can better focus on global contextual semantics, including important objects in the foreground and background contextual contents. Additionally, by visualizing filtered layer results of fine-grained ALF based on the VLP model – METER (Dou et al., 2022b) for NLVR and VQA tasks, we note varying patterns in the layers omitted by the model across different tasks. The reason may be that the simple shorter texts in VQA allow it to filter more fine-grained layers of the text tower compared to the long text with complex semantics in the NLVR task. Overall, visualizations clearly demonstrated the effectiveness of efficient filtering based on the proposed Double-Filter approach at both data- and network-level.

## 5. Conclusion

In this paper, we introduce a novel Double-Filter method for efficient fine-tuning of VLP models, aimed at optimizing both data input and network architecture. For data input optimization, we developed a new Image Patch Filter (IPF) that employs a pre-trained ViT model to identify significant patches in the foreground and background, detected by YOLO. This effectively reduces data redundancy while preserving comprehensive information. For network architecture optimization, we propose a fine-grained Architecture Layer Filter (ALF) based on an adjustable genetic algorithm (AGA). AGA efficiently eliminates redundant layers in the original VLP models through multi-generation crossover inheritance and mutation, enhancing the model's efficiency.

Extensive experimental evaluations on three downstream tasks, VQA , NLVR[2] and Cross-modal Retrieval, demonstrate that our Double-Filter method achieves a better balance between efficiency and performance in fine-tuning various VLP models.

**Limitation**. Our Double-Filter leverages patch importance weights derived from Vision Transformers (ViT) and selected based on a pre-trained YOLO detector. While they are widely adopted, this reliance may limit flexibility. In future work, we plan to explore alternative strategies for determining patch importance and patch selection, such as offline lightweight pre-trained models. In addition, we will use a better genetic algorithm variant to search for the optimal subnetwork structure to solve the problems of high computational complexity and easy convergence to local optimality during training.

## Impact Statement

This paper presents work whose goal is to advance the field of Machine Learning and Deep Learning. There are many potential societal consequences of our work, none which we feel must be specifically highlighted here.

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

# A. Appendix

## A.1. Hyperparameters and Details

In this section, we present the hyperparameters and implementation details of the proposed Double-Filter method, which consists of two main components: (1) Image Patch Filter (IPF) and (2) fine-grained Architecture Layer Filter (ALF).

### A.1.1. IMAGE PATCH FILTER (IPF):

For the METER model (Dou et al., 2022b), we assessed patch importance using the attention weights of the classification ([CLS]) token from the last layer of the pre-trained Vision Transformer (ViT) (Alexey, 2020). In the case of ViLT (Kim et al., 2021), all image patches were processed through the first self-attention layer, and the attention map of the first [CLS] token was extracted to evaluate patch importance. It is noteworthy that ViLT contains two [CLS] tokens: a text [CLS] token and a visual [CLS] token, the latter of which is used to separate modalities.

### A.1.2. ARCHITECTURE LAYER FILTER (ALF):

We focused on filtering the most computationally intensive components of each model. For METER (Dou et al., 2022b), this is the two-end fusion layer, and for ViLT (Kim et al., 2021), it is the transformer encoder.

ALF configurations were adjusted based on the specific requirements of each task. For the VQA task, we employed a population size of 100, 20 iterations, and a two-point crossover method. For the NLVR task, we used a population size of 50, 10 iterations, and a single-point crossover method. Loss calculations in the fitness function were based on 100 batches of training data (equivalent to 2 epochs), and losses were recorded over 10 validation batches.

After determining the optimal network structure, we applied the filtered network structure, followed by 10 epochs of training in combination with the IPF approach. The first epoch served as a warm-up, and after completing all the epochs, the model's performance was evaluated on the test set. For the VQA task, evaluations were conducted via an online platform[4].

During training, only the parameters of the additional modules, classifiers, pooler layers, modal-type embeddings, and adapters were updated; the rest of the network remained frozen. We adopted hyperparameters consistent with those in the original papers, and FLOPs calculations included YOLO detection to ensure comprehensive and accurate results.

All experiments were conducted on an RTX 3090 GPU. During evaluation, inference time was measured with a

---

[4]https://eval.ai/web/challenges/challenge-page/830/overview

---

*Table 5.* Experimental Parameters

| Common Parameters | |
| --- | --- |
| **Parameter** | **Value** |
| Training epoch | 10 |
| Warm-up epoch | 1 |
| Learning rate | $1 \times 10^{-5}$ |
| Adapter hidden dimension | 96 |

| Image Patch Filter (IPF) | | |
| --- | --- | --- |
| **Parameter** | **METER** | **ViLT** |
| CLS tokens | Last layer of ViT | First layer of ViLT |

| Architecture Layer Filter (ALF) | | |
| --- | --- | --- |
| **Parameter** | **VQA Experiment** | **NLVR Experiment** |
| Population size | 100 | 50 |
| Number of iterations | 20 | 10 |
| Crossover method | Two-point | Single-point |
| Training data per generation | 100 batches over 2 epochs | |
| Validation data per generation | 10 batches | |

batch size of 64 for METER and 128 for ViLT. Additional details on the Double-Filter method and hyperparameter settings are provided in Table 5.

## A.2. More Visualization

To more fully demonstrate our proposed Image Patch Filter (IPF) of the proposed novel **Double-Filter** for VLP models, we provide more detailed visualizations of the IPF results in Figure 5 and Figure 6 from the NLVR task and VQA [5] task. respectively. Through specific text questions, we

---

[5]Since we use online evaluation to evaluate the effectiveness of the VQA task, we cannot get an explicit answer. Therefore, we use the training examples for visualization of our IPF, which do not affect the effectiveness proof of IPF because it has no trainable parameters.

found that only by combining the context of important objects in the image and the background environment where they are located, the model can answer the questions more accurately. Compared to EViT (Liang et al., 2022), which focuses only on salient objects, our IPF can better focus on global contextual semantics, including important objects in the foreground and background contextual content, while effectively preserving key information in the image. This can effectively solve the corresponding text questions, especially those that require attention to background knowledge.

## A.3. Memory Usage

We conducted a comparative analysis of memory usage during inference across different fine-tuning methods, including our own model (Table 6). All evaluations were performed

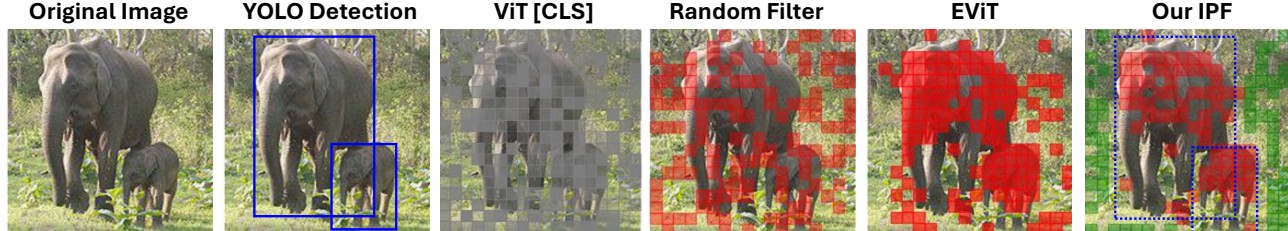

**Question: A baby elephant is standing in the grass next to an adult elephant without tusks.  Answer: True.**

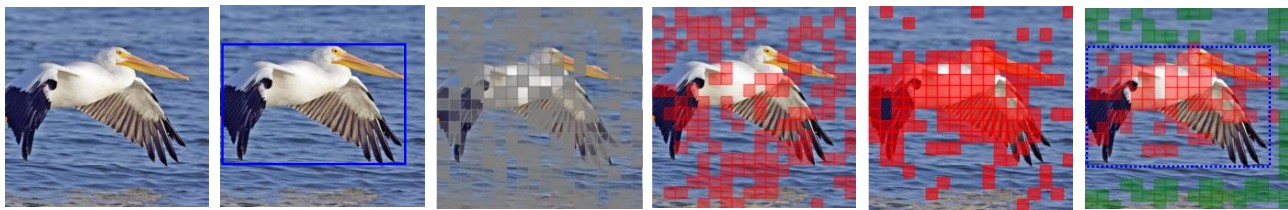

**Question: A bird files right above the water in the image on the right.  Answer: True.**

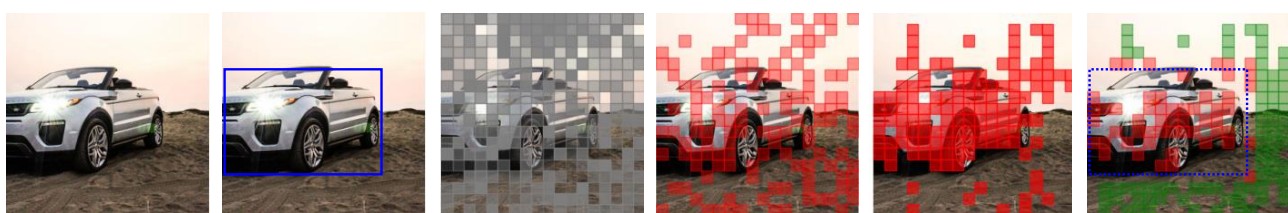

**Question: A convertible is in a parking space overlooking the beach.  Answer: True.**

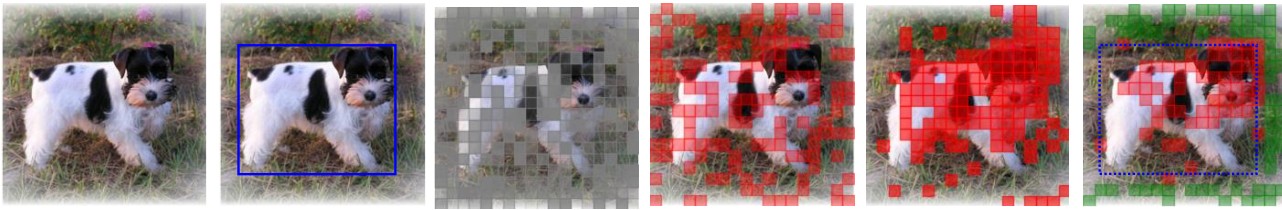

**Question: A black and white dog is standing in the grass looking at the camera.  Answer: True.**

*Figure 5.* Visualizations of the comparisons of different patch filters, including the Random Filter, EViT (Liang et al., 2022) and our proposed Image Patch Filter (IPF) in Double-Filter for METER (Dou et al., 2022b) on NLVR[2] test set for NLVR task.

with a batch size of 1 to ensure consistency.

| METER | Full tuning | Adapter | DAS | Double-Filter |
|-------|-------------|---------|-------|---------------|
| VQA | 3068M | 3090M | 2950M | **2906M** |
| NLVR | 3030M | 3050M | 2916M | **2884M** |

*Table 6.* Comparison of memory usage on two tasks with different tuning methods.

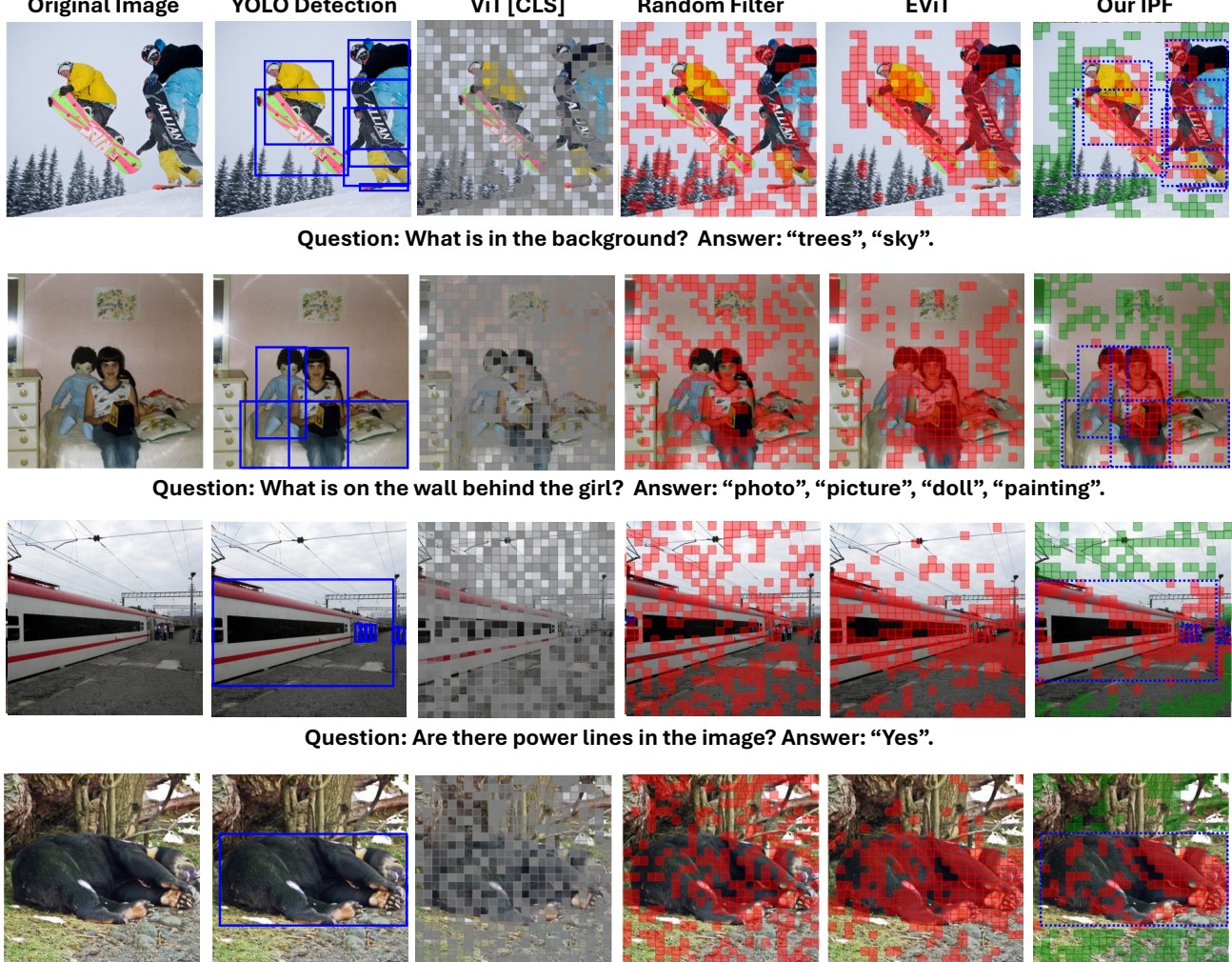

*Figure 6.* Visualizations of the comparisons of different patch filters, including the Random Filter, EViT (Liang et al., 2022) and our proposed Image Patch Filter (IPF) in Double-Filter for METER (Dou et al., 2022b) on VQA task. Note that, the VQA online test set does not provide standard answers, so we use the training examples to demonstrate the effectiveness of our IPF. Since IPF does not contain training parameters, it does not affect the effectiveness verification.

