# OpenReview forum: "Double-Filter: Efficient Fine-tuning of Pre-trained Vision-Language Models via Patch&Layer Filtering"
_ICML.cc/2025/Conference — ICML 2025 poster_

### Official Review · Reviewer_YEqs · 2025-03-08

**Overall Recommendation:** 4

**Summary:**

This paper presents Double-Filter, a method aimed at optimizing the fine-tuning process of vision-language pre-trained (VLP) models. The method reduces redundancy through two strategies: first, a novel patch selection approach that enhances feature representation via background-foreground separation; second, a genetic algorithm to eliminate redundant architectural layers, improving model efficiency.
Experiments on benchmark tasks, including VQA and NLVR2, with METER and ViLT models, show that Double-Filter achieves substantial reductions in computational complexity (e.g., up to 21.18 GFLOPs in VQA with METER) while maintaining competitive accuracy, with minimal performance degradation (0.27% on METER and 0.65% on ViLT). Visualizations further confirm that IPF effectively preserves global contextual semantics, and ALF ensures task-specific layer optimization.

## update after rebuttal
Authors' rebuttal have solved my concerns. I think previous score is high enough and I will keep my rating.

**Claims And Evidence:**

The claims made in the submission are clear and the paper provided enough evidence. The proposed method achieved significant reductions in FLOPs while maintaining competitive performance.

**Essential References Not Discussed:**

References are enough in this paper.

**Experimental Designs Or Analyses:**

1. It achieved significant reductions in FLOPs while maintaining competitive performance.

2. The ablation experiment is well designed and sufficient, clearly demonstrating the effectiveness of each key component proposed.

3. Clear visualization results are given to prove the effectiveness of the method.

**Methods And Evaluation Criteria:**

The proposed method combined Image Patch Filter (IPF) for data input optimization and Architecture Layer Filter (ALF) for network architecture optimization, providing a comprehensive solution for efficient fine-tuning. The motivation for fine-grained model redundancy removal is reasonable.
Experiment results show that the proposed method can reduce a large portion of input tokens while maintaining the performance.

**Other Comments Or Suggestions:**

A minor but important point: I noticed that the paper inconsistently uses "FLOPS" and "FLOPs" to refer to floating-point operations. It would be helpful to standardize the terminology throughout the paper for clarity and accuracy.

**Other Strengths And Weaknesses:**

Strength：
1. This paper appears to maintain a coherent and complete structure suitable for submission to ICML2025.

2. Achieves significant reductions in FLOPs while maintaining competitive performance. The topic of model architecture pruning is interesting and will be important for model deployment (e.g. on edge devices).

3. Unlike traditional methods that focus on coarse-grained block-level pruning, the ALF component employs a fine-grained filtering strategy. By targeting specific sub-layers (e.g., Multi-Head Self-Attention, Feed-Forward Network) within Transformer blocks, it achieves a more precise balance between model compression and performance retention.

Weakness：
1. The Image Patch Filter (IPF) relies on YOLO and ViT to determine patch importance. However, in complex scenarios or cases with occlusions, YOLO's object detection may be inaccurate, leading to incorrect foreground-background segmentation and potentially affecting the patch selection process.

2. The paper lacks a detailed analysis of YOLO's parameter count and FLOPs, which is important for assessing its computational cost in the proposed method.

3. Why does the paper not consider addressing redundancy in the text modality？

4. A minor but important point: I noticed that the paper inconsistently uses "FLOPS" and "FLOPs" to refer to floating-point operations. It would be helpful to standardize the terminology throughout the paper for clarity and accuracy.

**Questions For Authors:**

1. The Image Patch Filter (IPF) relies on YOLO and ViT to determine patch importance. However, in complex scenarios or cases with occlusions, YOLO's object detection may be inaccurate, leading to incorrect foreground-background segmentation and potentially affecting the patch selection process.

2. The paper lacks a detailed analysis of YOLO's parameter count and FLOPs, which is important for assessing its computational cost in the proposed method.

3. Why does the paper not consider addressing redundancy in the text modality？

**Relation To Broader Scientific Literature:**

The motivation for fine-grained model redundancy removal is reasonable. It maintain a coherent and complete structure suitable for submission to ICML2025.

**Theoretical Claims:**

This paper provided enough details in ensuring the correctness of its theoretical proofs. The author not only provides a clear description of each component of the model design but also presents well-defined assumptions, verifying the key efficiency design derivations and proof steps.

---

> ### Author Rebuttal · Authors · 2025-03-31
>
> We thank the reviewer for their very detailed and instructive feedback. We are very happy that you recognize our motivation and work and give positive support. Regarding your suggestions and concerns, we will respond one by one below, hoping to answer your questions.
>
> >_Q1. “YOLO's object detection may be inaccurate, leading to incorrect foreground-background segmentation and potentially affecting the patch selection process.”_
>
> Thanks for your comment.
> Indeed, the situation you mentioned is possible. But this is one of the motivations behind our decision to remove redundancy in the foreground and background separately and retain a certain proportion of patches respectively. On the one hand, our attention mechanism is based on the global attention distribution, so even if there are missed or wrong detections in the foreground, our IPF method can retain these important entity areas in the background (we assume they are important). Therefore, in this case, it can maintain the same global salient object attention ability as EViT(Liangetal.,2022). On the other hand, since we distinguish the foreground and background, we can pay more attention to the background than EViT, which only focuses on the salient area, even if there is no clear semantics, it is very important for the logical integrity of the image context expression.
>
> >_Q2. “The paper lacks a detailed analysis of YOLO's parameter count and FLOPs, which is important for assessing its computational cost in the proposed method.”_
>
> Thank you for the helpful suggestion. Specifically, YOLOv8 used in our IPF module requires approximately 4.5G FLOPs per forward pass on one RTX 3090Ti. Taking the METER model as an example, it requires a total of about 100G FLOPs cost in per forward pass, of which each transformer Block in the interaction network requires approximately 5G FLOPs. Therefore, YOLOs accounts for less than 5% of the total, which is approximately equal to the cost of one Block. We will revise Section 3.3 and Table 1 in the new paper to explicitly include this small cost to provide a complete picture of all forward computations.
>
> >_Q3. “Why does the paper not consider addressing redundancy in the text modality?”_
>
> Thank you for your insightful question and we appreciate the opportunity to clarify this aspect of our work.
> Our study focuses primarily on image redundancy because images account for the majority of computational overhead in Vision-Language Pretraining (VLP) models. **As discussed in Section 3.3 (Lines 259–264) of our paper**, textual redundancy is considerably lower than that of image patches, making image redundancy the more critical bottleneck to address.
> Tasks commonly used in VLP, such as image captioning, VQA, and cross-modal retrieval,etc., typically involve short text sequences. For example, in the METER and ViLT models we adopt, the maximum text length is only 40–50 tokens, whereas ViLT-B/32 processes over 240 image patches—meaning text accounts for less than 20% of the total input length. Similar observations are also reported in [1, 2].
> Moreover, since the computational cost (FLOPs) of transformer-based models scales linearly with sequence length, the relatively short length of text implies that the FLOPs consumed by the text modality are inherently much lower. Therefore, we concentrate on optimizing the visual modality, where redundancy is more significant and computational savings are more impactful.
>
> References:
>
> [1] Yang, Senqiao, et al. "Visionzip: Longer is better but not necessary in vision language models." arXiv preprint arXiv:2412.04467 (2024).
>
> [2] Kim, Wonjae, Bokyung Son, and Ildoo Kim. "ViLT: Vision-and-language transformer without convolution or region supervision." International Conference on Machine Learning, PMLR, 2021.
>
> >_Q4．A minor but important point: I noticed that the paper inconsistently uses "FLOPS" and "FLOPs" to refer to floating-point operations._
>
> Thank you for pointing this out. We will standardize the terminology throughout the paper, consistently using "FLOPs" to ensure clarity and accuracy. And we will carefully proofread the full text to ensure there are no typos.
>
>
> **We hope that the above responses could address the reviewers' concerns and questions.**

---

> > ### Comment · Reviewer_YEqs · 2025-04-02
> >
> > Thanks for the author's reply, which solved my confusions. In particular, the authors further gave explanations of the detailed computational cost of YOLO and the impact of the foreground and background on the model fine-tuning.
> >
> > Also, I browsed the comments of other reviewers and the author's reply, and found that the author supplemented the generalization experimental results on more complex retrieval task. As I commented before, I think the experimental setup of this paper, especially the ablation studies, is relatively complete.
> >
> > Overall, I'd support the full exploration of traditional machine learning methods to promote contributions to existing deep learning models. So I'd be willing to this paper.

---

> > > ### Author Response · Authors · 2025-04-02
> > >
> > > We sincerely thank the reviewers. Your recognition and support are the greatest encouragement for us to continue to improve our work.
> > >
> > > The reviewers clearly saw the core of our paper. Our motivation is indeed to further explore the innovation of traditional machine learning in efficient fine-tuning of pre-trained VLP models. We believe that this is in line with the theme of the ICML conference and to make extensive explorations for innovative applications of machine learning technology.

---

### Official Review · Reviewer_H3jf · 2025-03-13

**Overall Recommendation:** 3

**Summary:**

The paper introduces Double-Filter, a novel approach for efficient fine-tuning of vision-language pre-trained (VLP) models. The method addresses redundancy at both data and model levels. At the data level, an Image Patch Filter (IPF) leveraging a YOLO detector and ViT attention scores to retain only the most informative image patches is proposed, while at the model level, a fine-grained Architecture Layer Filter (ALF) using an adjustable genetic algorithm (AGA) to selectively replace redundant sub-layers within transformer blocks is designed. Experiments on VQA and NLVR tasks using the METER and ViLT models demonstrate that Double-Filter can significantly reduce FLOPs with only marginal performance degradation, thereby offering a compelling efficiency-performance trade-off.

## update after rebuttal
Thank you to the authors for their rebuttal, which has addressed most of my concerns. Hence, I maintain my recommendation of weak accept.

**Claims And Evidence:**

The paper makes several key claims: (1) The proposed dual filtering strategy effectively reduces both data-level and architecture-level redundancies, which in turn cuts down computational costs during fine-tuning. This is supported by theoretical FLOPs analysis and extensive ablation studies. (2) The Image Patch Filter preserves global semantic integrity by separately considering foreground and background regions, rather than focusing solely on salient objects. (3) The fine-grained Architecture Layer Filter, implemented via an adjustable genetic algorithm, enables selective pruning of sub-layers while retaining model performance. Experimental results comparing Double-Filter with other parameter-efficient fine-tuning (PEFT) methods (e.g., LoRA, Adapter, DAS) provide quantitative evidence for these claims, the visualization of IPF outputs supports the claim (2)

**Essential References Not Discussed:**

The paper comprehensively discussed the related works.

**Experimental Designs Or Analyses:**

Experiments are carried out on two widely adopted VLP models—METER and ViLT—across the VQA and NLVR benchmarks. The results indicate that Double-Filter achieves a substantial reduction in FLOPs (e.g., over 21G reduction for VQA on METER) while incurring only a minimal drop in accuracy. Detailed ablation studies further explore the impact of varying the patch filtering ratio and the number of layers pruned, validating the effectiveness of both the IPF and ALF components. The inclusion of inference speed comparisons also proves the effectiveness of proposed method in boosting VLP finetune.

**Methods And Evaluation Criteria:**

The methodology is built on two complementary components. The IPF first segments an image using a YOLO detector into foreground and background, then ranks patches based on [CLS] attention scores from a pre-trained ViT model. The ALF formulates layer pruning as an optimization problem solved by a genetic algorithm that iteratively adjusts a binary chromosome representation of the transformer’s sub-layers. Evaluation is conducted on two downstream tasks (VQA and NLVR) using common metrics such as accuracy and FLOPs reduction, alongside detailed ablation studies that examine varying patch filtering ratios and layer removal extents.

**Other Comments Or Suggestions:**

It would be beneficial for the authors to expand the evaluation to include a broader set of tasks or datasets could further validate the robustness of the approach. A discussion on hyperparameter sensitivity for the genetic algorithm would also help in understanding the stability of the ALF component.

**Other Strengths And Weaknesses:**

Strengths of the paper include its dual-level approach to reducing redundancy, a sound theoretical grounding through FLOPs analysis, and extensive experimental validation across multiple benchmarks. The modular design of the proposed filters offers flexibility and could be adapted to various VLP models. On the other hand, potential weaknesses include an increased system complexity due to the integration of a YOLO detector and genetic algorithm, and limited exploration of the method’s generalizability to tasks beyond VQA and NLVR.

**Questions For Authors:**

Could you elaborate on your choice of a genetic algorithm for the fine-grained Architecture Layer Filter? Given that GAs are heuristic and do not always guarantee an optimal result and requires hyperparameter tuning, did you consider alternative optimization methods that might offer stronger guarantees on the quality or stability of the optimization outcome?

**Relation To Broader Scientific Literature:**

The work builds on recent advances in efficient fine-tuning methods for VLP models, extending beyond traditional adapter-based approaches and block-skipping strategies (e.g., DAS(Wu
et al., 2024b)). The work also builds on a patch filtering techique that assigns an importance score to
each patch by leveraging the classification token ([CLS]) of the pre-trained ViT following (Liang et al., 2022).

**Theoretical Claims:**

The paper provides theoretical analysis regarding the reduction of FLOPs, demonstrating how the Double-Filter framework scales down the computational complexity of transformer blocks from the vanilla model. It also argues that selectively replacing sub-components (rather than entire blocks) offers a finer control over efficiency.However, the current experimental setup does not include a dedicated ablation study that directly compares selective sub-component replacement with whole block removal. A focused comparison—using image patch filtering (IPF) while contrasting ALF with block-level filtering methods would provide more definitive evidence for the benefits of fine-grained architectural control.

---

> ### Author Rebuttal · Authors · 2025-04-01
>
> We thank the Reviewer very much for the kind words, for the interest in our research activities, and for the very insightful comments.
>
> >_W1: “Potential weaknesses include an increased system complexity due to the integration of a YOLO detector and genetic algorithm, and limited exploration of the method’s generalizability beyond VQA and NLVR.”_
>
> Thanks for the comment.
> For the system complexity of the integration of a YOLO detector and genetic algorithm, as shown in Table 1 and Table 2, after introducing YOLO-based IPF and AGA-based ALF, our Double-Filter has significantly reduced the calculation of FLPOs, and the inference time has been greatly improved. Although the model introduces more algorithms, the inference efficiency of the VLP model has been significantly improved.
>
> For the exploration of the method’s generalizability to more tasks, we further experimented on more complex image and text retrieval tasks by renting a larger GPU. Due to time limits, the response Table reports our Double-Filter with same setting as Table 1 in the paper compared to the mainstream methods.
>
> METER    |  IR/TRR@1 | Add FLOPs    | ViLT    |  IR/TRR@1 | Add FLOPs
> -------------------|------------------|------------------|-------------------|------------------|------------------
> ClassifierOnly | 78.80/89.00  | 0.00      | - | 57.42/78.00  | 0.00
> Shallow Prompt| 74.20/88.60| +28.71G.   | -| 55.92/74.80| +19.53G
> Deep Prompt| 78.84/89.40  | +6.53G    | -| 58.64/79.50  | +5.14G
> LoRA      | 79.86/92.60   |0.00.      | -| 57.44/77.70   |0.00
> Adapter    | 80.38/91.90  |+1.64G.    | -| 62.68/81.40  |+0.86G
> Scaled PA   |  80.40/93.20 | +1.12G.  | -| 61.88/79.00 | +0.44G
> DAS       | 80.12/91.80.  |-11.16G  | - | 60.66/80.80  | -1.03G
> **Double-Filter** | 80.05/91.22  | -21.18G  | - | 61.18/79.39 | -4.72G
>
> On the cross-modal retrieval task, our proposed Double-Filter also shows the lowest computational complexity among existing PEFT methods, and maintains competitive performance. This further demonstrates the task generalization ability of our proposed Double-Filter method.
>
>
> >_Q1: “Could you elaborate on your choice of a genetic algorithm for the fine-grained Architecture Layer Filter?”_
>
> Thanks. **As description in Section 3.2, we propose an Adjustable Genetic Algorithm (AGA) for fine-grained architecture filtering, approximating an optimal configuration.** Based on the traditional genetic algorithm, it introduced adaptive adjustment for custom number of filtering layers, so as to better perform adaptive training and deployment capabilities on different devices. Each model configuration is encoded as a chromosome, with genetic operations (fitness evaluation, crossover, and mutation) guiding optimization across generations. The fitness function balances loss on sampled datasets, ensuring optimal layer reduction. Crossover swaps genes between parents, while mutation adjusts gene values to maintain a fixed number of replaced layers.
>
> Additionally, as mentioned in **Appendix A.1.2 (Lines 551-557), our loss calculations in the fitness function were based on 100 batches of training data (equivalent to 2 epochs), and losses were recorded over 10 validation batches**. The search cost of GA is equivalent to adding 2 epochs to the training phase, but DAS requires 3 epochs and requires full validation set validation. So the GA cost is still more efficient than the comparison method.
>
> Moreover, **please refer to our response to Q2, which further discusses optimization quality and stability.**
>
> >_Q2: “Given that GAs are heuristic and do not always guarantee an optimal result and requires hyperparameter tuning, did you consider alternative optimization methods that might offer stronger guarantees on the quality or stability of the optimization outcome?”_
>
> Thanks for the insightful comment. **As we said in Footnote2 on Page 5, identifying an absolute optimal structure is NP-hard, rendering exhaustive search impractical. So as mentioned in Section 3.2 (line 252), "The AGA aims to identify the approximate optimal reduction with L filtered layers."**
>
> Comparing with other searching algorithms, genetic algorithm uses population search to avoid falling into the local optimal solution, and has a greater probability to find the global optimal solution. And another reason we choose GA-based model is that genetic algorithms can be well parallelized and improve computational efficiency, because of the independent evaluation among individuals.
>
> To offer stronger guarantees on the quality or stability of the optimization outcome, the "Elite Strategy" is adopted in step 11 of Algorithm 2 to ensure that the optimal solution of the current generation is not lost during each generation update, thereby ensuring the quality of the search. We design the mutation operation to satisfy the constraint of the genetic algorithm while diversifying each generation effectively.
>
>
> **We hope that the above responses could address the reviewers' concerns and questions.**

---

### Official Review · Reviewer_pyf1 · 2025-03-13

**Overall Recommendation:** 5

**Summary:**

The paper introduces Double-Filter for refining the fine-tuning process of VLP models. It employs two key strategies to reduce redundancy: A new patch selection method that uses background-foreground separation to improve image feature selection; Then, a genetic algorithm designed to remove redundant architectural layers, thereby enhancing both the efficiency and effectiveness of the model. Together, these strategies aim to streamline the fine-tuning process while maintaining the model’s performance. Experimental results demonstrate that the proposed approach achieves competitive performance, with only marginal degradation compared to existing parameter-efficient fine-tuning methods, while significantly reducing computational complexity through the filtering of over 60% of image patches and 12 model layers.

**Claims And Evidence:**

Experimental results demonstrate that the proposed approach achieves competitive performance, with only marginal degradation compared to existing parameter-efficient fine-tuning methods, while significantly reducing computational complexity through the filtering of over 60% of image patches and 12 model layers.
The claims are clear with enough evidence.

**Essential References Not Discussed:**

Enough.

**Experimental Designs Or Analyses:**

Experimental results demonstrate that the proposed approach achieves competitive performance, with only marginal degradation compared to existing parameter-efficient fine-tuning methods, while significantly reducing computational complexity through the filtering of over 60% of image patches and 12 model layers.

The ablation experiments and visualization results are sufficient to demonstrate the effectiveness of each key component proposed.

**Methods And Evaluation Criteria:**

The proposed efficient training and inferencing Double Filter method makes it possible to deploy multimodal LLMs on low-resource devices by removing redundancy from both data and models, and also provides an effective reference for efficient training.

**Other Comments Or Suggestions:**

See Weaknesses.

**Other Strengths And Weaknesses:**

Strength：
1. The paper is well-written and easy to follow. And the motivation is reasonable

2. I appreciate that the authors provide a detailed proof process for the claimed efficiency while describing the model in detail (giving two detailed algorithmics). This paper provided enough details in ensuring the correctness of its theoretical proofs.

3. Double-Filter achieved significant reductions in FLOPs while maintaining competitive performance. Demonstrates versatility across multiple VLP models (METER and ViLT) and tasks (VQA and NLVR2).


Weakness：
1. The computational cost of YOLOv8 is included in Section 3.3 (FLOPs analysis). The authors claim that its cost is much lower than other modules, but no clear evidence is provided.
Please estimate the forward pass cost of this module, as it is part of the forward process of the proposed method. The same problem exists in Table 1, please consider all forward computations.

2. In the Image Patch Filter (IPF) section, the paper mentions sparsity ρ ∈ (0, 1). Is this sparsity a hyperparameter, and if so, how is it set or optimized during the training process? Could the authors provide more details on how this parameter is determined?

3. In the Fitness Function, why must \( \beta \) be greater than half of the maximum summed losses? How does this choice impact the optimization process?

4. Why choose METER and ViLT instead of other VLP models as the base models for experiments? The author needs to give a clearer explanation and reason.

5. On page 6, in the left column, there are two formulas that appear to be excessively long.

**Questions For Authors:**

1. In the Image Patch Filter (IPF) section, the paper mentions sparsity ρ ∈ (0, 1). Is this sparsity a hyperparameter, and if so, how is it set or optimized during the training process? Could the authors provide more details on how this parameter is determined?

2. Please estimate the forward pass cost of this module, as it is part of the forward process of the proposed method. The same problem exists in Table 1, please consider all forward computations.

3. Why choose METER and ViLT instead of other VLP models as the base models for experiments? The author needs to give a clearer explanation and reason.

4. In the Fitness Function, why must \( \beta \) be greater than half of the maximum summed losses? How does this choice impact the optimization process?

**Relation To Broader Scientific Literature:**

The research on VLP efficiency is very extensive. This paper introduces new thinking to this field through double filtering of data and model and based on the idea of ​​genetic algorithm, which is beneficial.

**Theoretical Claims:**

The theoretical method proposed in this paper is correct, and the detailed algorithms are given. In addition, this paper verifies the key efficiency design derivation.

---

> ### Author Rebuttal · Authors · 2025-04-01
>
> Thanks for your valuable and constructive comments. We appreciate your recognition of our work's innovative designs, evaluation criteria, and insightful analysis. Below, we address your concerns and questions individually:
>
> >_Q1: “In IPF section, the paper mentions sparsity ρ ∈ (0, 1). Is this sparsity a hyperparameter, and if so, how is it set or optimized during the training process? Could the authors provide more details on how this parameter is determined?”_
>
> Thanks for your detailed suggestion. The sparsity parameter (ρ) in the image patch filter is actually a hyperparameter that controls the filtering ratio of the image patches. The larger the sparsity parameter, the greater the filtering degree. In fact, in Table 3 of the paper, we have conducted ablation experiments by setting the sparsity parameter (ρ) differently to achieve a balance between efficiency and performance. We will further clarify this in the final version to avoid confusion.
>
> >_Q2: “The authors claim that YOLOv8’s cost is much lower than other modules, but no clear evidence is provided. Please estimate the forward pass cost of this module,...”_
>
> Thank you for the helpful suggestion. In our original analysis, we omitted the YOLOv8 cost because it is negligible compared to the overall model computation. However, for completeness, we now explicitly to calculate it. Specifically, YOLOv8 used in our IPF module requires approximately 4.5 G FLOPs per forward pass on one RTX 3090Ti. For the ViT-based encoder, taking the METER model as an example, it requires a total of about 100G FLOPs cost in per forward pass, of which each transformer Block in the interaction network requires approximately 5G FLOPs. Therefore, YOLOs accounts for less than 5% of the total, which is approximately equal to the cost of one Block.
>
> We will revise Section 3.3 and Table 1 to explicitly include this small cost to provide a complete picture of all forward computations.
>
> In addition, as shown in Table 1 and Table 3, even considering the cost of YOLO, our Double-Filter methods significantly reduces the computational cost when fine-tuning the VLP model, especially for models with complex interactions such as the METER model.
>
> >_Q3. “Why choose METER and ViLT instead of other VLP models as the base models for experiments? The author needs to give a clearer explanation and reason.”_
>
> Thanks for the insightful suggestion. Our choice of METER and ViLT as baselines was based on the following considerations:
>
> (1) METER and ViLT are the primary baselines for our comparison methods. To ensure a fair and meaningful evaluation, we followed the standard practice of using the same VLP models as our comparison baselines.
>
> (2) METER and ViLT represent two distinct VLP architectures. METER employs complex cross-modal fusion mechanisms, while ViLT maintains independent unimodal encoding. Our experimental results demonstrate that our Double-Filter achieves more significant efficiency improvements under the complex METER framework. This suggests that Double-Filter has the potential to be even more effective when applied to more complex VLP models.
>
> Given these considerations, we choose METER and ViLT. Nonetheless, we acknowledge the importance of further validation on additional VLP models and will explore this in future work.
>
>
> >_Q4. “In the Fitness Function, why must ( \beta ) be greater than half of the maximum summed losses? How does this choice impact the optimization process?”_
>
> Sorry for the confusion. In this paper, the role of $\beta$ is precisely to transform fitness values into the positive domain. In genetic algorithms, roulette wheel selection necessitates that the fitness scores generated by the fitness function be positive, as its core mechanism depends on transforming fitness values into non-negative and normalized probability distributions. If fitness values are negative, it not only leads to calculated selection probabilities being negative, which contradicts the fundamental principles of probability, but may also trigger division-by-zero errors or render probability calculations invalid when the sum of fitness values is zero or negative. Consequently, by employing a translation method to shift all fitness values into the positive domain, the relative ranking of individuals is maintained, thereby ensuring the algorithm's effective and reliable operation.  We will clearly these discussion in the new revision.
>
> >_Q5. “On page 6, in the left column, there are two formulas that appear to be excessively long.”_
>
> Thank you for the detailed suggestion! We'll simplify or break these formulas down into clearer segments to enhance readability.

---

> > ### Comment · Reviewer_pyf1 · 2025-04-04
> >
> > Thanks for the author's clarification, especially the further explanation of model choices and detail settings. Because our team is also concerned about the exploration of VLP efficient fine-tuning methods, and explored related efficient redundancy removal methods. The proposed Double-Filter is consistent with the experimental phenomenon we explored before, and the introduction of genetic algorithms to this task is eye-opening and inspiring to me.
> >
> > In addition, the FLOPs analysis in section 3.3 is useful to me, and I plan to continue to follow up the authors' work, especially after they clarify the details again in rebuttal. I think it's more complete.
> >
> > Although the authors had previously given detailed performance and efficiency comparisons, comprehensive results are further given during rebuttal, so I decided to further raise the score.

---

> > > ### Author Response · Authors · 2025-04-04
> > >
> > > Thank you for your thoughtful feedback and for sharing insights from your own research on VLP efficient fine-tuning and redundancy removal. We're glad to hear that our Double-Filter approach aligns with your experimental observations and that the integration of genetic algorithms was inspiring to you.
> > >
> > > We also appreciate your recognition of the FLOPs analysis in Section 3.3 and your continued interest in our work. Your constructive discussion during the rebuttal process helped us refine our explanations and make the paper more comprehensive.
> > >
> > > Thanks again for your support and for considering an updated score—it means a lot to us! We look forward to discussions in the future.

---

### Official Review · Reviewer_g8nL · 2025-03-13

**Overall Recommendation:** 3

**Summary:**

The paper introduces Double-Filter, an approach for efficient fine-tuning of Vision-Language Pre-trained (VLP) models by addressing redundancies at both the data and architectural levels. The authors propose two main components: (1) an Image Patch Filter (IPF) that selectively filters image patches by distinguishing between foreground and background regions, and (2) a Fine-grained Architecture Layer Filter (ALF) that employs an adjustable genetic algorithm (AGA) to identify and remove redundant sub-layers within transformer blocks. The authors claim that Double-Filter significantly reduces computational costs (e.g., 21.18G FLOPs reduction on METER) while maintaining competitive performance. The paper provides experimental results on METER and ViLT models, demonstrating the effectiveness of the proposed method in balancing efficiency and performance.

**Claims And Evidence:**

The claims made in the paper are generally supported by experimental evidence, but there are some areas where the evidence could be stronger or more comprehensive. The authors provide extensive ablation studies and comparisons with state-of-the-art PEFT methods, demonstrating that Double-Filter achieves significant reductions in computational costs (FLOPs) while maintaining competitive accuracy on VQA and NLVR tasks. For example, the paper reports that Double-Filter reduces FLOPs by 21.18G on the METER model for the VQA task, with only a minimal drop in accuracy (0.27% on average). However, the paper lacks a thorough discussion of the limitations of the proposed method. For instance, while the results are impressive on METER and ViLT, it is unclear how Double-Filter would perform on other VLP models, such as CLIP or ALIGN, which have different architectures and training objectives. Additionally, the paper does not explore the impact of patch filtering or layer removal on more complex multimodal tasks beyond VQA and NLVR, which limits the generalizability of the claims.

**Essential References Not Discussed:**

The paper covers most of the essential references related to efficient fine-tuning of VLP models. The paper could reference more recent advancements in genetic algorithms for neural architecture search, which might provide further context for the proposed AGA-based layer filtering approach.

**Experimental Designs Or Analyses:**

The experimental design is sound generally but there are some areas where the analysis could be improved. The authors conduct extensive experiments on two widely used VLP models (METER and ViLT) and compare their method against several state-of-the-art PEFT methods, including Shallow Prompt, Deep Prompt, LoRA, Adapter, and DAS. The ablation studies on patch filtering and layer removal are particularly insightful, demonstrating the impact of different filtering ratios and layer removal strategies on model performance and efficiency.

However, the experiments are limited to two downstream tasks (VQA and NLVR), which may not fully capture the generalizability of the proposed method. The authors do not explore the performance of Double-Filter on other tasks, such as image captioning or text-to-image generation, which are also common applications of VLP models. Additionally, the experiments are conducted on relatively small datasets (e.g., VQA 2.0), and it is unclear how the method would perform on larger or more diverse datasets.

Another limitation is the lack of analysis on the impact of Double-Filter on inference time and memory usage. While the paper focuses on FLOPs reduction, these metrics are critical for real-world applications, especially in resource-constrained environments.

**Methods And Evaluation Criteria:**

The proposed methods are partially suited for the problem of efficient fine-tuning of VLP models,  there are some areas where the approach could be improved. The Image Patch Filter (IPF) is a logical approach to reducing redundancy in image patches, which are known to be a major bottleneck in VLP models. By using a YOLO detector to separate foreground and background regions and then applying ViT attention scores to rank patch importance, the authors ensure that the most semantically relevant patches are retained. This approach is particularly effective for tasks like VQA and NLVR, where both foreground objects and background context are important.

However, the reliance on YOLO for foreground-background separation could be a limitation. YOLO is a computationally expensive object detection model, and its use adds overhead to the overall pipeline. The authors do not explore alternative object detection methods, which could potentially offer a better trade-off between accuracy and computational cost. Additionally, the patch filtering process is based on ViT attention scores, which may not always capture the full semantic importance of patches, especially in complex scenes with multiple objects or ambiguous backgrounds.

The Fine-grained Architecture Layer Filter (ALF) is a novel contribution, but the use of a genetic algorithm (GA) for layer removal introduces some challenges. While the GA is effective in exploring the space of possible layer configurations, it is computationally expensive and requires multiple generations of evaluation to converge. The authors do not discuss the computational cost of running the GA, which could be significant, especially for larger models or more complex tasks. Furthermore, the GA-based approach may not scale well to models with a large number of layers, as the search space grows exponentially with the number of layers.

**Other Comments Or Suggestions:**

N/A

**Other Strengths And Weaknesses:**

Strengths:

- The paper presents an effective approach to efficient fine-tuning of VLP models by addressing both data-level and architecture-level redundancies.

- The ablation studies and visualizations provide valuable insights into the impact of patch filtering and layer removal on model performance and efficiency.

Weaknesses:

- The paper could benefit from a more detailed discussion of the limitations of the proposed method, particularly in scenarios where patch filtering or layer removal might negatively impact performance.

- The authors could explore additional methods for determining patch importance, beyond the ViT-based approach, to further improve the robustness of the IPF.

- The reliance on YOLO for foreground-background separation adds computational overhead, and alternative object detection methods should be explored.

**Questions For Authors:**

1. The paper mentions that the proposed method is effective under METER and ViLT models. Have the authors tested Double-Filter on other VLP models.
2. The paper focuses on reducing FLOPs and maintaining accuracy. Have the authors considered the impact of Double-Filter on inference time and memory usage.
3. The paper uses a YOLO detector for foreground-background separation. Have the authors explored other object detection methods.

**Relation To Broader Scientific Literature:**

The paper is situated within the broader literature on efficient fine-tuning of VLP models. The authors discuss related work on adapter-based methods, block-skipping strategies, and other PEFT approaches, highlighting how their method differs by addressing both data-level and architecture-level redundancies. For example, the authors compare their approach to DAS, which also aims to reduce architectural redundancy but does so at a coarser level by replacing entire transformer blocks.

**Theoretical Claims:**

The paper does not present any theoretical proofs, so there are no theoretical claims to evaluate. The focus is primarily on empirical results and algorithmic contributions. However, the authors do provide a detailed analysis of the FLOPs reduction achieved by Double-Filter. The paper could benefit from a more rigorous theoretical analysis of the proposed methods. For example, the authors could provide a theoretical justification for why certain layers are more redundant than others, or why the proposed patch filtering strategy is optimal for preserving semantic information.

---

> ### Author Rebuttal · Authors · 2025-04-01
>
> We thank the reviewer for the thorough review, acknowledging our contributions, and providing instructive suggestions. We respond to each weakness (_W*_) or question (_Q*_) below.
>
> >_W1: “... explore the impact on more complex multimodal tasks beyond VQA and NLVR, ...”_
>
> Thanks for the instructive suggestion. We conducted VQA and NLVR experiments on one NVIDIA RTX 3090Ti, but the retrieval task requires many negative samples, making it difficult to maintain the same batch size compared to existing methods.
>
> To address the concerns, we rented a larger GPU and conducted generalization experiments on Flickr30K for image-text retrieval. The response Table reports our Double-Filter with same setting as Table 1 in the paper compared to the mainstream methods.
>
> METER    |  IR/TRR@1 | Add FLOPs    | ViLT    |  IR/TRR@1 | Add FLOPs
> -------------------|------------------|------------------|-------------------|------------------|------------------
> ClassifierOnly | 78.80/89.00  | 0.00      | - | 57.42/78.00  | 0.00
> Shallow Prompt| 74.20/88.60| +28.71G.   | -| 55.92/74.80| +19.53G
> Deep Prompt| 78.84/89.40  | +6.53G    | -| 58.64/79.50  | +5.14G
> LoRA      | 79.86/92.60   |0.00.      | -| 57.44/77.70   |0.00
> Adapter    | 80.38/91.90  |+1.64G.    | -| 62.68/81.40  |+0.86G
> Scaled PA   |  80.40/93.20 | +1.12G.  | -| 61.88/79.00 | +0.44G
> DAS       | 80.12/91.80.  |-11.16G  | - | 60.66/80.80  | -1.03G
> **Double-Filter** | 80.05/91.22  | -21.18G  | - | 61.18/79.39 | -4.72G
>
> Our Double-Filter achieves the lowest computational complexity among PEFT methods and maintains competitive performance in image-text retrieval.
>
> >_W2: “The authors do not discuss the computational cost of running the GA, which ...”_
>
> Thanks for the detailed suggestion. As mentioned in **Appendix A.1.2 (Lines 551-557), our loss calculations in the fitness function were based on 100 batches of training data (equivalent to 2 epochs), and losses were recorded over 10 validation batches**. The search cost of GA is equivalent to adding 2 epochs to the training phase, but DAS takes 3 epochs and requires full validation set validation. So the GA cost is still lower than the comparison method.
>
> >_W3: “A more detailed discussion of the limitations of the proposed method.”_
>
> Thanks for the detailed suggestion. We will further discuss detailed limitations. On the one hand, our method relies on pre-trained target detectors, but it still adds additional inference time; on the other hand, we use genetic algorithms to search for the best subnet, which has a long search time in the training stage and is prone to fall into the limitation of local optima. We will add the discussion in the final version.
>
> >_Q1: “Have the authors tested Double-Filter on other VLP models.”_
>
> Thanks for the instructive comment. We choose METER and ViLT as baselines because:
>
> (1) METER and ViLT are the primary baselines for our comparison methods. To ensure a fair evaluation, we followed the standard practice of using the same VLP models as our baselines.
>
> (2) METER and ViLT represent two mainstream VLP architectures, and others generally follow these architectures. METER employs complex cross-modal fusion, while ViLT maintains independent unimodal encoding. Our results show that Double-Filter achieves notable efficiency improvements, especially under METER, suggesting its potential for more complex VLP models.
>
> >_Q2: “Have the authors considered the impact of Double-Filter on inference time and memory usage.”_
>
> Thanks. In fact, **as shown in Table 2 (Line 332), we have compared the inference speed of Double-Filter with other methods**. Our results demonstrate that Double-Filter enables processing more samples per unit time, indicating a clear efficiency improvement. The following table provides additional memory consumption (inferencing with batch size of 1) to further illustrate the lower memory of Double-Filter.
>
> METER     | Full tuning  |  Adapter    | DAS       | **Double-Filter**
> ------------------|------------------|------------------|-------------------|-------------------
> VQA       |   3068M  | 3090M     |  2950M    | **2906M**
> NLVR      |   3030M   | 3050M    | 2916M      | **2884M**
>
>
> >_Q3: “Have the authors explored other object detection methods.”_
>
> Thank you for highlighting this vital direction.  We employed YOLOv8 because the YOLOs are the most mainstream object detectors, which can better balance effectiveness and efficiency of detection. We further test EfficientDet[1], which need more computing resources. Due to the limited rebuttal time, we only compared the IPF filtering overlap among detectors, and found that over 96% of filtered patches remained consistent, so we believe that this has little impact on the final patch retention. Future work will explore alternative methods, including lightweight offline pre-training.
>
> [1] Tan, Mingxing, Ruoming Pang, and Quoc V. Le. "EfficientDet:Scalable and efficient object detection." CVPR. 2020.

---

### Official Review · Reviewer_Qv8A · 2025-03-14

**Overall Recommendation:** 3

**Summary:**

The paper proposes Double-Filter, an efficient fine-tuning framework for vision-language pre-trained (VLP) models. It combines two redundancy reduction techniques: (1) an Image Patch Filter (IPF) that leverages YOLO for foreground/background separation and a ViT-based [CLS] attention mechanism to retain informative image patches, and (2) a fine-grained Architecture Layer Filter (ALF) using a genetic algorithm to prune redundant transformer sub-layers. The method is evaluated on visual question answering (VQA) and natural language visual reasoning (NLVR) tasks with METER and ViLT models, showing reduced computational costs (FLOPs) with minimal performance degradation compared to baseline parameter-efficient methods.

**Claims And Evidence:**

The paper claims Double-Filter reduces FLOPs by 21.18G (METER/VQA) and 12.51G (METER/NLVR) while maintaining accuracy within 0.5% of full fine-tuning. However, the inclusion of YOLO's computational cost is not accounted for in efficiency metrics , potentially overstating FLOP reductions. The experimental FLOP analysis excludes the YOLO component (Figure 2, Section 3.3), which may add significant overhead during inference, undermining the claimed efficiency.

**Essential References Not Discussed:**

Not found.

**Experimental Designs Or Analyses:**

1. Evaluation Scope:
The experiments focus on two tasks (VQA 2.0 and NLVR2) with METER and ViLT as baselines. While these are standard benchmarks, the exclusion of retrieval tasks (e.g., COCO evaluation) limits the validity of claims about broad efficiency. VLP models are often evaluated across a more diverse set of tasks, so the absence of retrieval results weakens the argument for general utility.

2. FLOP Calculation:
The FLOP analysis excludes YOLO’s computational cost, which could be significant. The claimed 21.18G FLOP reduction for METER/VQA may be offset by YOLO’s overhead, especially during inference. The analysis assumes constant FLOPs for tasks with different input sizes (e.g., VQA’s image + text vs. NLVR’s dual images + text), which may not reflect real-world scenarios.

**Methods And Evaluation Criteria:**

The proposed methods (Double-Filter) and evaluation criteria (VQA/NLVR benchmarks with METER/ViLT) align with the objective of improving fine-tuning efficiency for VLP models. However, the evaluation scope is narrow, excluding retrieval tasks critical for VLP benchmarking. YOLO’s overhead in the IPF step is also not accounted for, which could skew FLOP reduction claims.

**Other Comments Or Suggestions:**

N/A

**Other Strengths And Weaknesses:**

Strengths:

This paper is well written and organized.

Additional Weaknesses:

The experimental design lacks diversity, focusing solely on METER/ViLT with fixed sparisty ratios (e.g., 60% patch filtering). Scalability to larger VLP models (e.g., OFA, BEIT-3) or datasets (COCO, Conceptual Captions) is untested.

The supplementary material (Appendix) provides incomplete details on genetic algorithm parameters or YOLO hyperparameters, limiting clarity on implementation.

Incorporating YOLO adds complexity and potential latency to the pipeline, contradicting the efficiency goals. YOLO’s inference time likely outweighs the savings from patch pruning, especially in real-time applications.

The Architecture Layer Filter (ALF) draws parallels to LayerDrop (Fan et al., 2019) and DAS (Wu et al., 2024b), which prune entire transformer blocks. ALF’s fine-grained pruning (targeting MHSA/MHCA/FFN sub-layers) is more granular, akin to methods like Dynamic Head Pruning (Michel et al., 2019). The use of a genetic algorithm for layer selection is novel but conceptually related to AutoML frameworks (e.g., NAS), though this paper doesn’t position ALF as a neural architecture search. The lack of benchmarking against LayerDrop-style dynamic inference suggests a gap in comparing to adaptive methods.

**Questions For Authors:**

How does YOLO’s computational cost compare to the FLOP savings from patch/layer pruning? Can the pipeline function without YOLO for fair comparison to prior work?

Why were retrieval tasks excluded, given their importance for VLP benchmarking?

**Relation To Broader Scientific Literature:**

It combines two key techniques: (1) an Image Patch Filter (IPF)[1] that uses YOLO for foreground/background separation and ViT [CLS] tokens to retain informative patches, and (2) a fine-grained Architecture Layer Filter (ALF) employing a genetic algorithm to prune redundant transformer sub-layers.

[1] Liang, Youwei, et al. "Not all patches are what you need: Expediting vision transformers via token reorganizations." arXiv preprint arXiv:2202.07800 (2022).

**Theoretical Claims:**

The only theoretical component is the FLOP analysis. The derivation of FLOPs for transformer components (MHSA, MHCA, FFN) is mathematically sound, using standard matrix multiplication cost formulas. However, the analysis oversimplifies assumptions (e.g., ignoring tokenization/embedding layers, assuming perfect parallelism). YOLO’s FLOPs are excluded, making the theoretical cost reductions optimistic.

Other claims (e.g., redundancy removal effectiveness) are empirical and lack theoretical backing.

---

> ### Author Rebuttal · Authors · 2025-04-01
>
> Thanks for the encouraging feedback and suggestions. However, we argue that some concerns were addressed in our paper, so we clarify them again here. We address the main weaknesses(W*) and questions(Q*) below:
>
> >_W1: “YOLO's computational cost is not accounted for in efficiency metrics ,....”_
>
> **Clarification:**  We would like to clarify that our paper explicitly states that our experiments take YOLO’s FLOPs into account when calculating the model FLOPs,  **in the corresponding captions of Table 1 (Line 278) and Table 3(Line 332), e.g. “_The FLOPs of Double-Filter contain the YOLO detector._”.**
> We compare complete Double-Filter with others in Table 1 and show the FLOPs reduction statistics via the diversity IPF in Table 3.
> We hope this clarification resolves the reviewer’s concern.
>
> >_W2: “The experimental design lacks diversity, focusing solely with fixed sparsity ratios (e.g., 60% patch filtering).”_
>
> First, we clarify that Table 3 and Figure 3 present ablation studies on various sparsity ratios, which is inconsistent with the reviewer's claim of a fixed ratio.
>
> Second, exhaustively testing all ratio combinations would demand substantial computational resources. To balance performance and efficiency, we evaluate four representative ratios—40%, 50%, 60%, and 70%. This approach ensures a comprehensive yet feasible analysis and follows a widely adopted experimental methodology.
>
> >_W3: “YOLO’s inference time likely outweighs the savings from patch pruning.”_
>
> As shown in Table 2, YOLO’s inference time is included in Double-Filter. Despite this, it still improves overall inference speed. We will clarify this in the final version.
>
> >_W4: “The Appendix provides incomplete details...”_
>
> The main parameters of the genetic algorithm were given. We guess the concerns lie in YOLO. We clarify that our pipeline does not rely on additional settings of YOLO, and directly uses the pre-trained YOLOv8 (Reisetal.,2023). Therefore, it does not affect our reproducibility. We will add more details in the final version.
>
> >_W5: “The lack of benchmarking against LayerDrop-style dynamic inference ...”_
>
> LayerDrop-style methods typically required modifications during the pre-training stage to enable adaptive inference, whereas our focus lies in fine-tuning pre-trained VLP models without altering the upstream pre-training process. This makes the comparison less aligned in terms of scope—note that DAS (Wu et al., 2023) also does not compare with such methods. Moreover, unlike dynamic methods that often require loading the entire model and selectively dropping layers, our method only loads a searched subnetwork, reducing memory and computational overhead.
>
> >_Q1: “How does YOLO’s computational cost compare to the FLOP savings from patch/layer pruning?”_
>
> Taking METER as an example, as shown in Table 1, our Double-Filter can reduce computational costs by 21.18G FLOPs. The computational costs consider the YOLO’s FLOPs (about 4.5G FLOPs), so there are about 26G FLOPs savings from IPF and ALF. Additionally, Table 3 also shows the detailed computational costs including YOLO’s FLOPs when applying different filter ratios for IPF. The FLOPs saved by image patch filtering far exceeds the introduction of YOLO.
>
> >_Q2: “Can the pipeline function without YOLO for fair comparison to prior work?”_
>
> YOLO effectively distinguishes foreground and background without complex design, enhancing versatility. In future work, we will further explore how to replace YOLO, but this will undoubtedly require a tailored design, and the versatility may be reduced. Additionally, as we clarified in W1, our DF included YOLO's cost when comparing with others, so they are fair comparisons.
>
> >_Q3: “Why were retrieval tasks excluded, given their importance for VLP benchmarking?”_
>
> Sorry for the confusion. Due to resource limitations, we conducted VQA and NLVR experiments on an NVIDIA RTX 3090Ti. However, retrieval require numerous negative samples, making it difficult to maintain the same batch size as in prior works (e.g., DAS used NVIDIA Tesla A100 GPU).
>
> To address the concerns, we rented a larger GPU for image-text retrieval task on Flickr30K.
>
> METER    |  IR/TRR@1 | Add FLOPs    | ViLT    |  IR/TRR@1 | Add FLOPs
> -------------------|------------------|------------------|-------------------|-----------------|-----------------
> ClassifierOnly | 78.80/89.00  | 0.00      | - | 57.42/78.00  | 0.00
> Shallow Prompt| 74.20/88.60| +28.71G.   | -| 55.92/74.80| +19.53G
> Deep Prompt| 78.84/89.40  | +6.53G    | -| 58.64/79.50  | +5.14G
> LoRA      | 79.86/92.60   |0.00.      | -| 57.44/77.70   |0.00
> Adapter    | 80.38/91.90  |+1.64G.    | -| 62.68/81.40  |+0.86G
> Scaled PA   |  80.40/93.20 | +1.12G.  | -| 61.88/79.00 | +0.44G
> DAS       | 80.12/91.80.  |-11.16G  | - | 60.66/80.80  | -1.03G
> **Double-Filter** | 80.05/91.22  | -21.18G  | - | 61.18/79.39 | -4.72G
>
> Our model for retrieval task still maintains a high efficiency while ensuring the performance with the same setting as Table 1.

---

### Decision · Program_Chairs · 2025-05-01

**Decision:**

Accept (poster)

**Comment:**

This paper proposes a double filter approach to compress the fine-tuning process of vision-language pre-trained (VLP) models via filtering redundancies in feature inputs and architectural components. A patch selection and redundant layer elimination are developed for performance preservation and model acceleration. It received positive reviews mostly. The reviewers are generally acknowledge the contribution of the proposed method. One reviewer [Qv8A] mentions the experimental issue regarding the comparison diversity, which is solved via the author responses. Overall, the AC has checked all the files, and feels the current work is ready for reporting at present. The authors shall incorporate reviewers' comments in the camera-ready revisions.